# CRISPR tiling deletion screens reveal functional enhancers and allelic compensation effects (ACE) on *SIN3A* transcription

Xingjie Ren[1], Lina Zheng[2], Yuxi Liu[1], Lenka Maliskova[1], Tsz Wai Tam[1], Yifan Sun[1], Hongjiang Liu [1], Xiekui Cui[1], Jerry Lee[1], Maya Asami Takagi[1], Bin Li[3], Bing Ren[3,4,5], Wei Wang [2,3,6] & Yin Shen [1,7,8] ✉

Precise transcriptional regulation is critical for cellular function and development, yet the mechanism of this process remains poorly understood for many genes. To gain a deeper understanding of the regulation of neuropsychiatric disease risk genes, we identify a total of 39 functional enhancers for four dosage-sensitive genes, *APP*, *FMR1*, *MECP2*, and *SIN3A*, using CRISPR tiling deletion screening in human induced pluripotent stem cell (iPSC)-induced excitatory neurons. More importantly, we discover that allelic enhancer deletions at *SIN3A* could be compensated by increased transcriptional activities from the other intact allele. Such allelic compensation effects (ACE) on transcription are stably maintained during differentiation and, once established, cannot be reversed by ectopic *SIN3A* expression. Further, ACE at *SIN3A* occurs through dosage sensing by the promoter. Together, our findings unravel a regulatory compensation mechanism that ensures stable and precise transcriptional output for *SIN3A*, and potentially other dosage-sensitive genes.

Optimal spatial-temporal gene regulation is pivotal to normal development. Mutations in *cis*-regulatory elements (CREs), such as enhancers, cause target gene misregulation and contribute to diseases[1,2]. To date, over one million candidate CREs (cCREs) have been mapped in the human genome based on biochemical signatures, including chromatin accessibility, histone modifications, and transcription factor (TF) binding sites[3,4]. cCREs are also enriched for variants identified by genome-wide association studies (GWAS) for complex diseases, signifying their potential contribution to human diseases through gene regulatory mechanisms[4]. However, how cCREs regulate target gene expression remains mostly uncharacterized.

Genetic analyses have identified numerous risk genes for neuropsychiatric and neurodegenerative diseases, many of which are dosage-sensitive genes[5], suggesting that precise regulation of gene expression is critical for maintaining normal neuronal function and preventing disease. For example, mutations and duplication in *APP*, a precursor protein of β-amyloid peptide[6] are causal factors in Alzheimer's disease (OMIM, 104300)[7]. Elevated *FMR1* transcription of *FMR1* premutations (55-200 CGG repeats at the 5' untranslated region) increases the risk of developing fragile X-associated tremor/ataxia syndrome (FXTAS) (OMIM, 300623), fragile X-associated primary ovarian insufficiency (FXPOI) (OMIM, 311360), and fragile X-associated

[1]Institute for Human Genetics, University of California, San Francisco, San Francisco, CA, USA. [2]Bioinformatics and Systems Biology Graduate Program, University of California, San Diego, La Jolla, CA, USA. [3]Department of Cellular and Molecular Medicine, University of California, San Diego, La Jolla, CA, USA. [4]Center for Epigenomics, University of California, San Diego, La Jolla, CA, USA. [5]Moores Cancer Center, University of California, San Diego, La Jolla, CA, USA. [6]Department of Chemistry and Biochemistry, University of California, San Diego, La Jolla, CA, USA. [7]Department of Neurology, University of California, San Francisco, San Francisco, CA, USA. [8]Weill Institute for Neurosciences, University of California, San Francisco, San Francisco, CA, USA. ✉e-mail: yin.shen@ucsf.edu

neuropsychiatric disorders (FXAND), while full mutations of *FMR1* (>200 CGG repeats) completely inhibit *FMR1* transcription, resulting in fragile X syndrome (OMIM, 300624)[8]. In another example of MeCP2, a methyl-CpG-binding protein[9], loss-of-function mutations in *MECP2* lead to Rett syndrome (OMIM, 312750)[10], and duplication of *MECP2* causes a neurodevelopmental disorder, *MECP2* duplication syndrome (OMIM, 300260)[11]. Finally, heterozygous loss-of-function variants in *SIN3A*, a transcriptional repressor[12], cause *SIN3A* haploinsufficiency, giving rise to neurodevelopmental syndromes including Witteveen-Kolk syndrome (OMIM, 613406) and Autism Spectrum Disorder (OMIM, 209850)[13,14]. These observations of disease conditions resulting from gene dosage alterations underscore the essential role of regulatory mechanisms in safeguarding the genome against deleterious mutations, thereby preventing pathological shifts in gene expression.

To better understand the gene regulatory program for those dosage-sensitive genes, we performed unbiased CRISPR tiling deletion screening of enhancers for *APP*, *FMR1*, *MECP2*, and *SIN3A* using CREST-seq (for *cis*-regulatory element scan by tiling-deletion and sequencing)[15] during the differentiation of human induced pluripotent stem cells (iPSCs) into excitatory neurons. Through extensive validation, we uncovered an unexpected transcriptional compensation mechanism that maintains the stable transcriptional output of *SIN3A* upon allelic enhancer deletions.

## Results

### Allelic tiling deletion CRISPR screens identify enhancers for neurological risk genes

To identify functional enhancers for *APP*, *FMR1*, *MECP2*, and *SIN3A* genes in neurons, we performed CREST-seq[15] for unbiased tiling deletion CRISPR screening of genomic sequences surrounding the gene of choice. These genes are strategically chosen due to their importance in both developmental and disease perspectives, as well as their involvement in pathogenesis linked to gene dosage alterations. Specifically, we generated allelically tagged EGFP or mCherry reporters in the WTC11 i³N iPSC line[16] to monitor allelic gene expression during the CRISPR screens using fluorescence-activated cell sorting (FACS) (Fig. 1a and Supplementary Fig. 1a–c). The WTC11 i³N iPSC line is a male cell line containing the integrated doxycycline-inducible *Ngn2* at the *AAVS1* locus, which allows us to generate a large quantity of homogeneous excitatory neurons[16] (Fig. 1a and Supplementary Fig. 1d). For *APP* and *SIN3A*, EGFP and mCherry are tagged on each allele, and for X-linked *FMR1* and *MECP2*, we tagged them with either a mCherry or an EGFP reporter, respectively (Fig. 1a and Supplementary Fig. 1b). We designed approximately 11,000 to 17,000 paired-guide RNAs (pgRNAs) targeting 2-4 Mbp around each gene. pgRNAs mediated deletions had an average size of 2000–3500 bp and 15× or 20× coverage for each nucleotide (Supplementary Fig. 2a–e). We infected each iPSC reporter line with the corresponding lentivirus library expressing

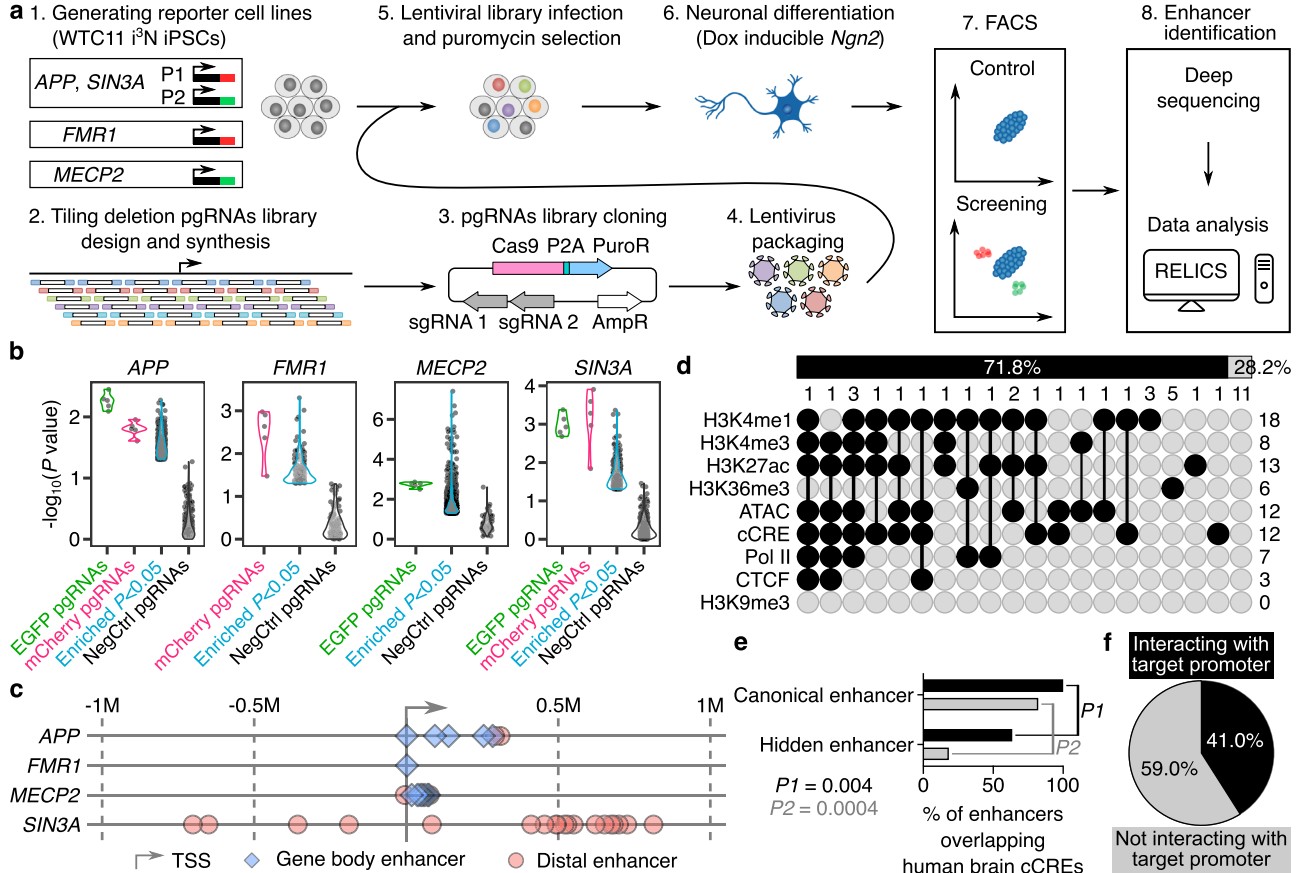

**Fig. 1 | Identification and analysis of enhancers of four genes. a** The workflow of identifying enhancers of *APP*, *FMR1*, *MECP2*, and *SIN3A* in iPSC-induced excitatory neurons using CRISPR tiling deletion screening. **b** Violin plots showing the distribution of *P* values for enriched pgRNAs (log₂FC > 0) in each screen. The positive control pgRNAs targeting EGFP and mCherry, and some of the pgRNAs targeting screening regions, are significantly enriched in each screen. The negative control pgRNAs are not significantly enriched. **c** The distribution of identified enhancers of *APP*, *FMR1*, *MECP2*, and *SIN3A*, relative to TSS of each target gene. **d** Upset plot

showing the overlap between identified enhancers and each chromatin feature. The numbers in each row and column indicate the total number of enhancers in each category. **e** Overlap between identified enhancers and cCREs identified from human brain samples. *P* values were determined using the two-tailed Fisher's exact test. Data from PMID: 38781369, 38092918, and 37824643 are labeled in gray, and data from PMID: 34663447 are labeled in black. **f** The percentage of enhancers interacting and not interacting with target promoters based on H3K4me3 PLAC-seq data. Source data are provided as a Source data file.

SpCas9 protein and pgRNAs, selected infected cells with puromycin for 1 week, and then differentiated iPSCs into excitatory neurons (Fig. 1a). Two weeks after differentiation, we sorted out neurons with reduced reporter expression using FACS (Supplementary Fig. 3a). For each gene, we performed two independent screens (Supplementary Fig. 3b, c). To assess the screening strategy, we quantified the frequency of pgRNAs in each sample and calculated the fold change in pgRNA counts between FACS-sorted cells and control cells. As expected, positive control pgRNAs targeting EGFP and mCherry were significantly enriched in FACS-sorted populations with reduced reporter expression, whereas non-targeting negative control pgRNAs showed no enrichment, validating our screening strategy (Fig. 1b and Supplementary Fig. 4a). Previous studies showed that the inhibition of *SIN3A* affects neuronal differentiation, whereas *APP*, *FMR1*, and *MECP2* do not[17,18]. These results highlight the strength of our screening strategy in identifying functional regulatory elements for both essential and non-essential genes.

Given that reduced reporter expression was used as the readout, we define the identified functional elements as enhancers. We identified 39 enhancers for 4 genes using RELICS[19] (Supplementary Fig. 4b, c and Supplementary Data 1). On average, these functional enhancers are 315.3 kb away from the transcriptional start sites (TSSs) of their target genes, with 16 enhancers located within their target gene bodies (Fig. 1c). As anticipated, none of the identified enhancers overlap with the repressive chromatin marker H3K9me3 (Fig. 1d). 71.8% (28/39) enhancers overlap with active chromatin signatures profiled in WTC11 i³N iPSC-derived excitatory neurons, including chromatin accessibility[17], H3K4me1, H3K4me3, H3K27ac, H3K36me3, and the binding of CTCF and RNA polymerase II[20], or cCREs annotated in excitatory neurons from the human brain samples[21] (Fig. 1d). Notably, 28.2% (11/39) of enhancers are not associated with the chromatin signatures of enhancers we examined. This is consistent with reports of the existence of hidden enhancers that do not have conventional chromatin marks for cCRE[15,22,23]. We further examined whether the hidden enhancers are cell type–specific by assessing their overlap with cCREs identified from human brain samples[21,24,25], and cCREs generated by a large meta-analysis of human brain epigenome data[26]. We found that, compared with canonical enhancers, hidden enhancers tend to remain hidden across multiple cell types (Fig. 1e). Interestingly, only 41.0% (16/39) of enhancers participate in H3K4me3 associated chromatin interactions[17] (Fig. 1f), confirming the notion that while chromatin interactions are valuable for delineating enhancer-promoter relationships, other mechanisms also play a role in enhancer-mediated transcriptional regulation[27,28].

### Functional validation of CREST-seq identified enhancers

We focused on validating enhancers located in gene bodies by examining their effects on target gene expression through CRISPR deletion followed by flow cytometry analysis (Supplementary Fig. 5a). For *FMR1*, deleting one enhancer (FMR1-E1, located in the first intron of *FMR1*) reduced expression of *FMR1-mCherry* in both iPSCs and excitatory neurons (Fig. 2a, b and Supplementary Figs. 5b–d and 6). For MECP2, deleting three MECP2 gene body enhancers, MECP2-E3, MECP2-E8, and MECP2-E10, led to the downregulation of *MECP2-EGFP* in both iPSCs and excitatory neurons, while deleting MECP2-E6 caused downregulation of *MECP2-EGFP* only in excitatory neurons (Fig. 2c and Supplementary Figs. 6 and 7a–c), suggesting MECP2-E6 is a neuron-specific enhancer. MECP2-E6 overlaps with enhancer associated chromatin markers such as H3K27ac and H3K4me1 in neurons, but not in iPSCs, which is consistent with our validation results (Supplementary Fig. 8a). The dependence of *MECP2* for the three shared enhancers was further confirmed with independent enhancer deletion clones (Fig. 2d). The reduction of *MECP2* transcription was more profound in clones with deletions of MECP2-promoter, MECP2-E8, and MECP2-E10 compared to MECP2-E6 (Fig. 2e), suggesting varied effects of

enhancers on *MECP2* expression. Deleting APP-E3, located in the last intron of *APP*, led to a similar downregulation of *APP* as deleting the *APP* promoter in both iPSCs and excitatory neurons (Fig. 2f and Supplementary Figs. 6 and 9a, b).

In addition to gene body enhancers, we validated a distal enhancer, SIN3A-E4, for *SIN3A*. After Cas9 and pgRNA delivery, a subpopulation of cells exhibited significant downregulation of *SIN3A-EGFP* or *SIN3A-mCherry* in both iPSCs and 2-week excitatory neurons, confirming that SIN3A-E4 is a functional enhancer of *SIN3A* (Fig. 2g). As expected, we only observed the deletion of SIN3A-E4 on one of the two alleles consistent with the fact that *SIN3A* is a haploinsufficient gene and an essential gene in neurons[13] (Supplementary Figs. 6 and 10a–d). Cells with further perturbation of the 19 bp CTCF motif in SIN3A-E4 located at a TAD boundary (chr15:74765001-74770000), exhibited reduced *SIN3A-EGFP* or *SIN3A-mCherry* expression (Fig. 2h, i). Genotyping of cells with reduced *SIN3A-EGFP* or *SIN3A-mCherry* expression revealed various deletions, insertions, and substitutions at the CTCF motif (Fig. 2j and Supplementary Fig. 11a), confirming the importance of the CTCF binding motif in the SIN3A-E4 enhancer.

### Allelic deletion of *SIN3A* enhancer triggers allelic compensation effects (ACE)

The dual reporter tagging of *SIN3A* enabled us to monitor the allelic *SIN3A* transcription followed by enhancer deletions. Remarkably, cells with reduced expression of *SIN3A-EGFP* exhibited increased expression of *SIN3A-mCherry*, and vice versa, upon deleting the SIN3A-E4 enhancer (Fig. 2g), suggesting that enhancer deletion on one allele induced allelic compensation effects (ACE) from the other allele. As *SIN3A* is a haploinsufficient gene, we hypothesize that allelic enhancer perturbation may trigger ACE to maintain a steady level of transcriptional output, which may serve as a crucial genome defense mechanism against deleterious non-coding mutations affecting *SIN3A* expression. To examine whether other enhancer deletions could similarly trigger ACE, we deleted another three *SIN3A* enhancers located in various genomic regions, SIN3A-E2 (*CYP1A1* intron), SIN3A-E3 (*CYP1A2* exons, *CYP1A2* is not expressed in neurons with RPKM = 0), and SIN3A-E5 (non-coding intergenic regions) (Fig. 3a). After the delivery of Cas9 and pgRNAs for deleting these enhancers, cells exhibited significant downregulation of either *SIN3A-EGFP* or *SIN3A-mCherry* expression, but elevated reporter expression on the other allele in both iPSCs and 2-week excitatory neurons (Fig. 3b, c and Supplementary Fig. 10a–d), confirming that ACE is a general mechanism of *SIN3A* transcriptional regulation.

Bona fide enhancers only affect transcription in *cis*. To ensure observed allelic gene expression changes are due to enhancer deletion in *cis*, we picked two phased SNPs in the WTC11 genome. The first SNP is located in the last intron of *SIN3A* (chr15: 75374632, C/T, hg38), which was used for resolving the allelic information of tagged EGFP and mCherry reporters. The second SNP is located adjacent to SIN3A-E2 (chr15: 74721849, T/G, hg38), which was used for the identification of the allele with the enhancer deletion. Our results showed that cells with allelic enhancer deletions have reduced SIN3A-EGFP or SIN3A-mCherry expression from the same allele (Supplementary Fig. 12a). Therefore, we demonstrate that the ACE arises from the opposite allele, compensating for reduced *SIN3A* transcription caused by the enhancer deletion in *cis*.

Enhancer deletion-induced ACE can also be further confirmed with allelic gene expression analysis leveraging an SNP in the *SIN3A* intron in the WTC11 iPSC genome (Fig. 3d, chr15: 75374632, C/T, hg38). In wild-type clones (G+M+), we observed a near 1:1 expression ratio from both alleles. However, clones with allelic enhancer deletion with either reduced EGFP expression (G-M+) or reduced mCherry expression (G+M-) exhibit dominant expression from either the C allele or the T allele, respectively, in both iPSCs and 2-week excitatory neurons (Fig. 3e). More importantly, the total *SIN3A* mRNA level remains no

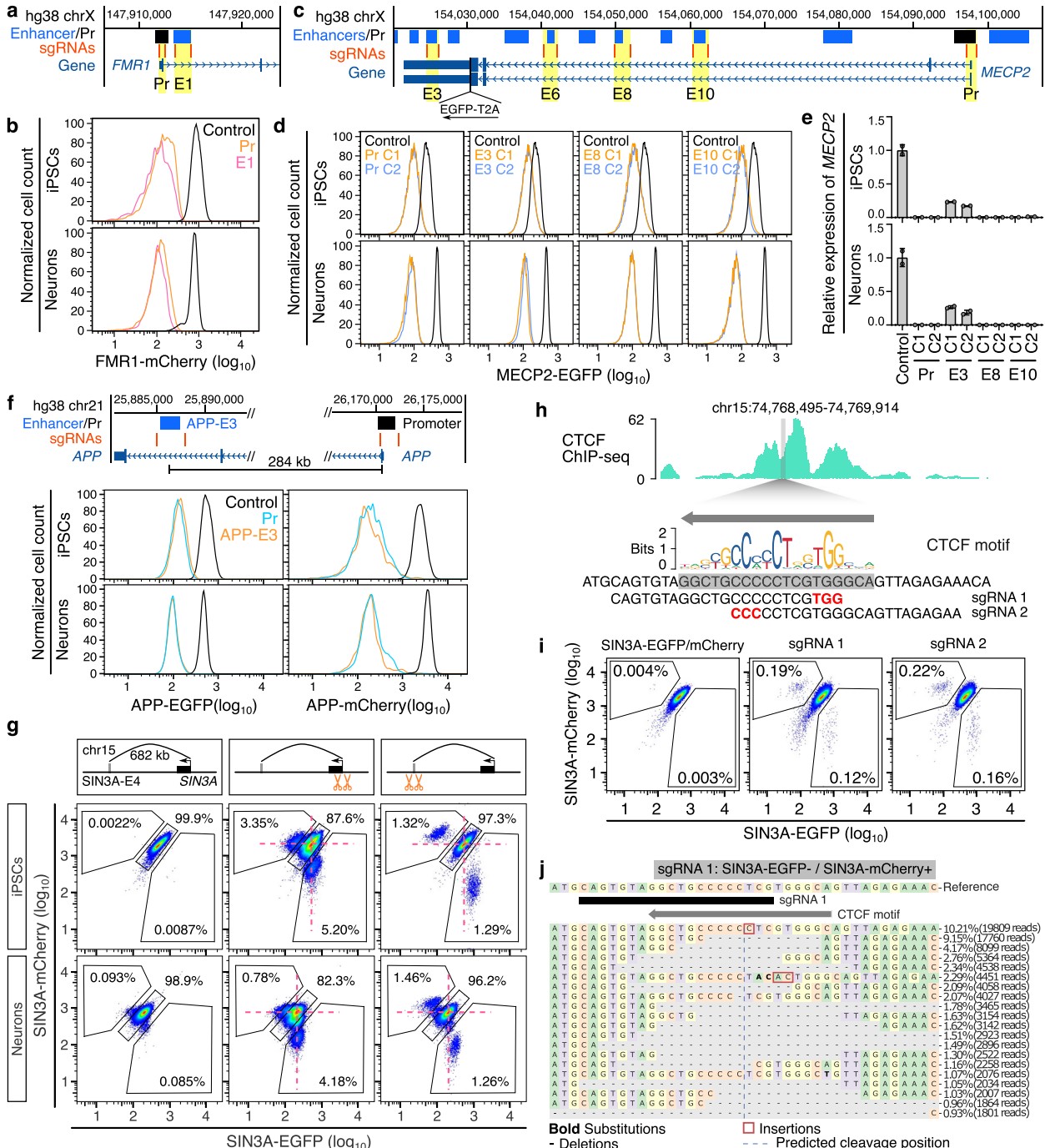

**Fig. 2 | Validating CREST-seq identified enhancers. a** Genome browser screenshot showing gene body enhancer of *FMR1* and sgRNAs targeting *FMR1* promoter and enhancer. **b** Flow cytometry plots showing the significant downregulation of FMR1-mCherry expression after deleting *FMR1* promoter and FMR1-E1 enhancer in both iPSCs and excitatory neurons. Positive controls (black line) are the FMR1-mCherry reporter cells. **c** Genome browser screenshot showing identified enhancers of *MECP2* and sgRNAs targeting *MECP2* promoter and enhancers. **d** Single clones of *MECP2* promoter or enhancers deletion showing significant downregulation of MECP2-EGFP in both iPSCs and excitatory neurons. Positive controls (black line) are the MECP2-EGFP cells. C1 and C2 indicate two independent clones. **e** RT-qPCR results showing the significant downregulation of *MECP2* expression in each clone ($P < 0.05$ for all the clones, two-tailed two-sample *t*-test; $n = 2$). Data are shown as mean ± SEM. **f** Flow cytometry plots showing the significant downregulation of APP-EGFP or APP-mCherry in *APP* promoter and APP-E3 deletion cells. Positive controls (black line) are the APP-EGFP/mCherry reporter cells. **g** Flow cytometry plots showing the downregulation of SIN3A-EGFP or SIN3A-mCherry in *SIN3A* promoter and SIN3A-E4 deletion cells. Red dashed lines indicate the position of SIN3A-EGFP/mCherry double-positive cells. **h** The genome browser screenshot showing the CTCF ChIP-seq signal in SIN3A-E4 enhancer region in WTC11 iPSCs. The CTCF motif was obtained from JASPAR. Two sgRNAs were designed to target the CTCF motif. PAM sequences were in red. **i** Flow cytometry plots showing the downregulation of SIN3A-EGFP or SIN3A-mCherry in sgRNA 1 and sgRNA 2 infected cells. **j** The editing outcomes of sgRNA 1 in the cells of SIN3A-EGFP-/SIN3A-mCherry +. Source data are provided as a Source data file.

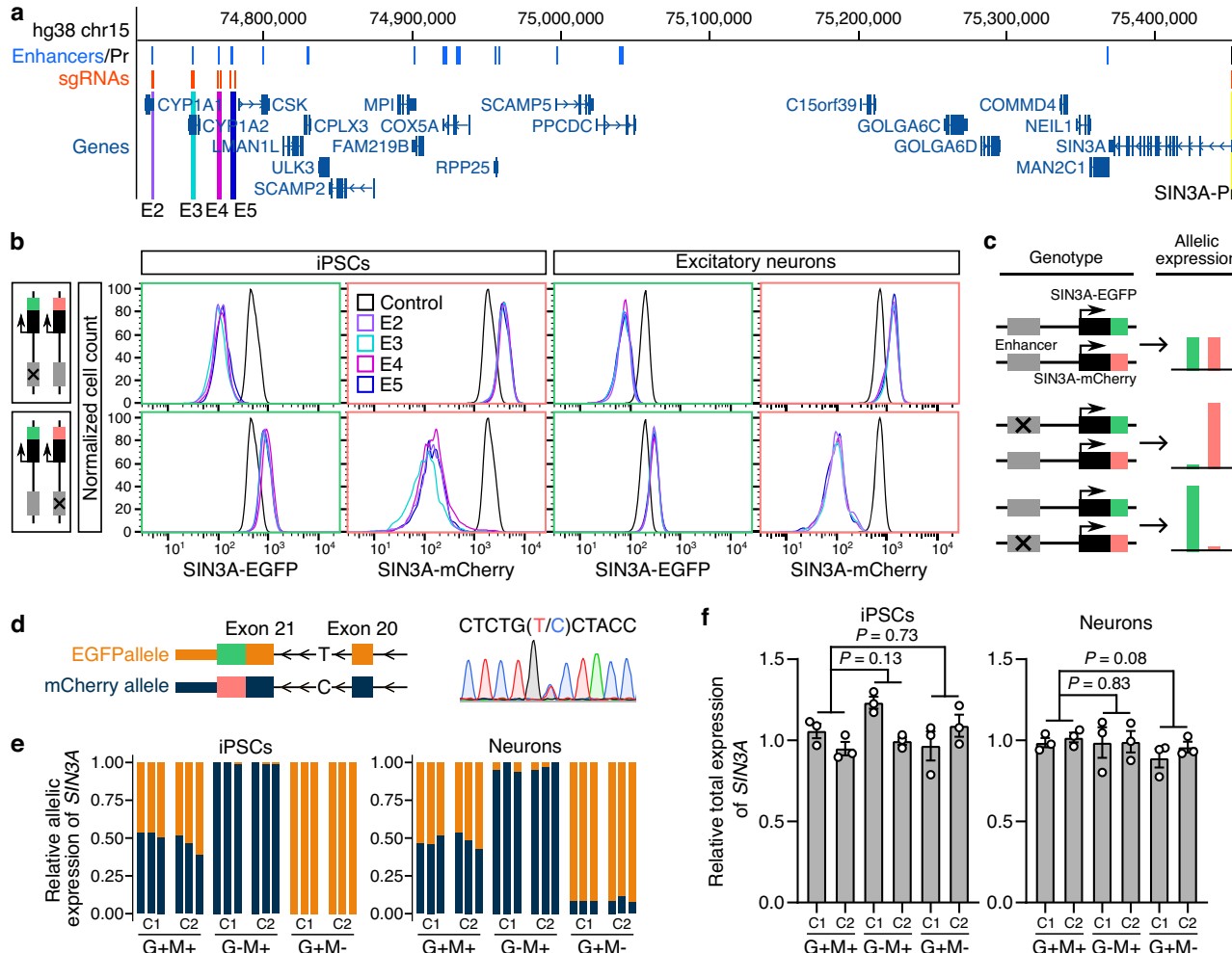

**Fig. 3 | Allelic enhancer deletion induces transcriptional compensation of SIN3A. a** Genome browser screenshot showing enhancers of *SIN3A* and sgRNAs targeting *SIN3A* promoter and enhancers. **b** Flow cytometry plots showing the significant downregulation of SIN3A-EGFP and SIN3A-mCherry expression after deleting *SIN3A* enhancers. Positive controls (black lines) are SIN3A-EGFP/mCherry reporter cells. **c** The model of the allelic expression pattern of *SIN3A* and the associated genotype. **d** Sanger sequencing shows the SNP in *SIN3A* intron. **e** Allelic gene expression analysis using the SNP located in *SIN3A* intron shows dominant expression from one allele in G-M+ (SIN3A-EGFP-/SIN3A-mCherry+) and G+M-

(SIN3A-EGFP+/SIN3A-mCherry-) clones in both iPSCs and 2-week excitatory neurons. Dark blue color indicates the C allele, and orange color indicates the T allele. **f** RT-qPCR results showing the total *SIN3A* expression in each clone relative to *GAPDH*. Data are shown as mean ± SEM. *P* values were determined using the two-tailed two-sample *t*-test. The results in (**e**, **f**) are from the analysis of the SIN3A-E4 enhancer. C1 and C2 represent two independent clones for each genotype, and each clone comprises three biological replicates. Source data are provided as a Source data file.

changes across all the clones (Fig. 3f), suggesting that ACE is used to achieve the precise transcriptional output of *SIN3A*.

To explore the mechanism of ACE in response to enhancer deletions, we performed a time course analysis of allelic expression changes upon deleting one enhancer (SIN3A-E4) and compared that to deleting *SIN3A* promoter in iPSCs. Cells with the reduced SIN3A-EGFP or SIN3A-mCherry signals appeared 2 days after the delivery of Cas9 and pgRNAs (Fig. 4a). To track the ACE, we quantified the SIN3A-EGFP and SIN3A-mCherry signals in cells with either the enhancer or the promoter deletion. In cells with the SIN3A-E4 enhancer deletion, the downregulation of either *SIN3A-EGFP* or *SIN3A-mCherry* allele is positively correlated with the upregulation of the other allele over time (Fig. 4b, c, R² = 0.92 for the EGFP allele, R² = 0.92 for the mCherry allele). In contrast, we only observed the downregulation of either *SIN3A-EGFP* or *SIN3A-mCherry* allele in cells with the promoter deletion without apparent ACE from the opposite allele (Fig. 4b, c, R² = 0.027 for the EGFP allele, R² = 0.08 for the mCherry allele). To check the kinetics of ACE from the enhancer deletion, we calculated the slope between each pair of adjacent time points. The absolute slope value

exceeded one after day 5, reached the summit at day 10, and dropped quickly at the end (Fig. 4d). The observed dynamic rate of ACE after enhancer deletion suggests that ACE is more potent as *SIN3A* expression approaches the level that triggers haploinsufficiency after day 5. In addition, the ACE rate decreases as the total *SIN3A* expression level approaches the wild-type level. In contrast, promoter deletion-induced SIN3A downregulation remained constant after day 5 (Fig. 4b, c). Long-read RNA-seq data revealed *SIN3A* transcription from two TSSs[29], and we only deleted the promoter of the major *SIN3A* transcript (TSS1 region, chr15:75450931-75452275) (Supplementary Fig. 13a, b). Thus, the partial reduction of *SIN3A* expression from the promoter deletion allele may not be sufficient to induce ACE. Indeed, after deleting *SIN3A* entire promoter regions (Deletion #1, chr15:75450928-75456304; Deletion #2, chr15:75450928-75456867), we did not observe ACE in these cells (Supplementary Fig. 13b, c). These results demonstrate that ACE is a dynamic process initiated from significantly reduced expression of *SIN3A* from one allele.

To explore whether the established ACE persists during neuronal differentiation, we isolated single clones either with no enhancer

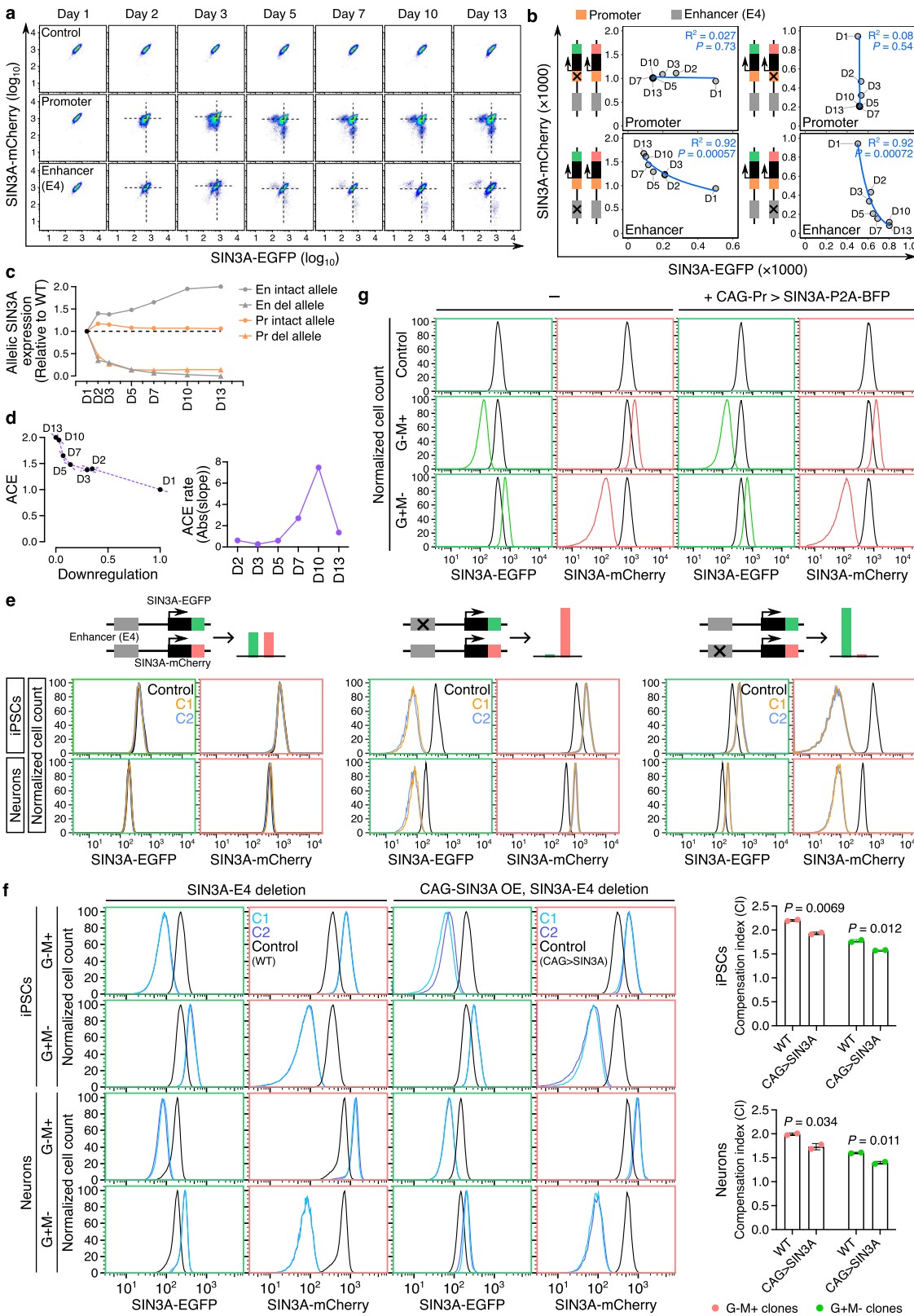

deletion (SIN3A-EGFP+/SIN3A-mCherry+: G+M+) or with allelic enhancer deletions (SIN3A-EGFP-/SIN3A-mCherry+: G-M+; SIN3A-EGFP+/SIN3A-mCherry-: G+M-). We observed that transcriptional compensation remains unchanged after differentiating iPSCs into excitatory neurons (Fig. 4e), suggesting that the ACE of *SIN3A*, once established, can be heritably maintained throughout the differentiation. Further, we tested whether the *SIN3A* overexpression can inhibit the ACE by

deleting SIN3A-E4 enhancer in *SIN3A* reporter cells with *SIN3A* over-expression (CAG>SIN3A, SIN3A-EGFP/mCherry) (Supplementary Fig. 13d). We isolated single clones with enhancer deletion and checked allelic *SIN3A* expression. By calculating the compensation index (CI), defined as the ratio of EGFP or mCherry signal (from the compensation allele) in enhancer deletion clones relative to control cells. We observed that *SIN3A* overexpression significantly reduced the

**Fig. 4 | Allelic enhancer deletion-induced allelic compensation effect (ACE) is a dynamic process. a** Flow cytometry plots showing the expression of SIN3A-EGFP and SIN3A-mCherry in control cells (SIN3A-EGFP/mCherry reporter cells) and cells infected with pgRNAs targeting *SIN3A* promoter and SIN3A-E4 enhancer. The dates refer to the days following the lentivirus infection. **b** Dot plots showing the expression trend of SIN3A-EGFP and SIN3A-mCherry signals in the cells with reduced expression level of SIN3A-EGFP or SIN3A-mCherry in (**a**). Trendlines are based on logarithmic model. **c** Allelic promoter and enhancer deletion-induced downregulation of *SIN3A*. Dots indicate the levels of SIN3A-EGFP or SIN3A-mCherry in cells with allelic promoter or enhancer deletions. The black dashed line indicates allelic expression levels from wild-type cells. **d** The ACE rate of *SIN3A* enhancer E4 deletion. The average downregulation and transcriptional compensation resulting from enhancer deletion on the EGFP and mCherry alleles were used to calculate the slope between each pair of adjacent time points. **e** Flow cytometry plots showing

the SIN3A-EGFP and SIN3A-mCherry signals from each clone in iPSCs and neurons. SIN3A-EGFP/mCherry reporter cells were used as control. C1 and C2 indicate two independent clones of each genotype. **f** The ACE in SIN3A-E4 deletion clones isolated from SIN3A-EGFP/mCherry reporter cells with and without CAG promoter-controlled ectopic *SIN3A* expression. C1 and C2 indicate two independent clones for each genotype. Control cells are SIN3A-EGFP/mCherry reporter cells, with (CAG>SIN3A) or without SIN3A overexpression (WT). The compensation index (CI) is defined as the ratio of EGFP or mCherry signal (from the compensation allele) in enhancer deletion clones relative to control cells. *P* values were determined using the two-tailed two-sample *t*-test. **g** Flow cytometry plots showing the SIN3A-EGFP and SIN3A-mCherry signals in control cells and SIN3A-E4 deletion clones with and without CAG promoter-controlled ectopic *SIN3A* expression. SIN3A-EGFP/mCherry reporter cells were used as control. Source data are provided as a Source data file.

CI in both iPSCs and neurons, compared with the CI in cells without *SIN3A* overexpression (Fig. 4f). These results indicate that exogenous *SIN3A* expression can suppress the ACE response.

Since ACE is triggered by allelic enhancer deletion-induced *SIN3A* downregulation, we wondered whether it can be reversed by elevating *SIN3A* expression. To test this, we ectopically expressed a *SIN3A* transgene driven by the *CAG* promoter and *SIN3A* promoter in SIN3A-E4 deletion clones, which resulted in more than 3-fold (*CAG* promoter) and about 1.7-fold (*SIN3A* promoter) expression of *SIN3A* compared to the endogenous expression level (Supplementary Fig. 13d, e). However, *SIN3A* overexpression is not sufficient for disrupting endogenous transcriptional compensation in SIN3A-E4 deletion clones (Fig. 4g, Supplementary Fig. 13f). These results suggest ACE, once established, can not be reversed by increasing *SIN3A* expression.

### The *SIN3A* promoter mediates allelic enhancer deletion-induced ACE

Next, we investigate how cells can sense reduced *SIN3A* expression upon enhancer deletion and initiate the process of ACE on transcription. As a transcriptional repressor, SIN3A binds to its own promoter[30], suggesting autoregulatory feedback (Supplementary Fig. 13b). This prompted us to consider that the *SIN3A* promoter could mediate SIN3A dosage sensing to achieve an optimal transcriptional level of the *SIN3A* gene. To test whether the promoter is responsible for initiating ACE, we tested the activities of two *SIN3A* promoter reporters (P1, P1 + P2) with and without shRNA-mediated downregulation of endogenous *SIN3A* expression. First, we showed that both P1 and P1 + P2 promoter reporters exhibit strong EGFP expression, confirming that they are active promoters (Fig. 5a and Supplementary Fig. 13b). Both P1 and P1 + P2 promoter reporters exhibited a significant increase of promoter activity when endogenous *SIN3A* expression is reduced by *SIN3A* shRNA (Fig. 5b–d). In addition, we confirmed the SIN3A binding at P1 and P2 promoter regions using ChIP-qPCR (Fig. 5e). These results suggest allelic enhancer deletion leads to near complete loss of *SIN3A* in *cis*, resulting in less *SIN3A* binding at its own promoters, which triggers ACE via the upregulating of SIN3A from the trans allele. In contrast, allelic partial deletion of promoter retained partial *SIN3A* expression in *cis* (Fig. 4c), which is not sufficient to trigger ACE (Fig. 5f). Our ACE model can also explain the haploinsufficiency of *SIN3A* for the Witteveen-Kolk syndrome (WITKOS) patients with large deletions of the entire *SIN3A* locus including the *SIN3A* promoter[13,31,32] (Supplementary Fig. 14a), while copy number loss variant (RCV000139947) overlapped with *SIN3A* enhancers identified from clinical samples is likely benign. In WITKOS patients, the loss of one copy of the promoter disrupts promoter-mediated SIN3A dosage sensing, resulting in only half of the normal expression of *SIN3A* from the intact wild-type allele. This reduced level of SIN3A is insufficient to support normal cellular function, leading to haploinsufficiency in WITKOS patients. Furthermore, our ACE model suggests that RCV001834362 is less likely

contribute to the disease, which overlaps with the SIN3A enhancer and is annotated as uncertain significance in the ClinVar database.

Leveraging the feature of protein binding to their own gene promoter, we matched protein-coding promoter sequences with the known TF binding motifs database[33,34] to identify promoters that could be bound by the TFs expressed from the same promoter. In total, we identified 530 human and 321 mouse TF genes with their promoters harboring their own binding motifs (Supplementary Fig. 15a). Gene ontology enrichment analysis for those genes yielded terms associated with transcriptional regulation, cis-regulatory region DNA binding, and nucleus localization, consistent with their roles as TFs (Supplementary Fig. 15b). Considering SIN3A is a transcriptional repressor, 279 human and 180 mouse repressive TFs could be subjected to enhancer deletion-induced ACE (Supplementary Fig. 15a). Leveraging RNA-seq data from human tissues in GTEx[35], we found that those 279 human genes are widely expressed across human tissues (Supplementary Fig. 15c). Since ACE is used to maintain the steady expression of associated genes, we further checked their dosage sensitivity using the ClinGen database with dosage-sensitive information for 1,545 genes[36] and a machine learning predicted genome wide gene dosage sensitivity map[5]. Among 279 genes, 45 were found in the ClinGen database, and 270 were found in the dosage sensitivity map. In both analyses, there is a significant enrichment of human candidate genes in haploinsufficiency, instead of triplosensitivity (84.4 vs. 4.4% in ClinGen, 47.7 vs. 25.2% in gene dosage sensitivity map) (Supplementary Fig. 15d). These candidate genes suggest that ACE could be a widespread gene regulatory mechanism for dosage-sensitive genes. Indeed, we found ACE happened in *NANOG* enhancer deletion clones in a recent study in hESCs[37] (Supplementary Fig. 16a, b), and *NANOG* is one of the candidate genes in our predicted gene list. The genes from our prediction are TFs, which drive precise transcription patterns[38], and are known to be enriched for haploinsufficient genes[39] with genetic studies highlighting the significance of their dosage for normal development[40,41].

## Discussion

In this study, we identified functional enhancers for four neuropsychiatric risk genes in iPSC-derived excitatory neurons using CREST-seq. Since *APP*, *FMR1*, *MECP2*, and *SIN3A* are dosage-sensitive genes associated with neuropsychiatric diseases, discovering their enhancers in neurons may offer new genomic loci for developing therapeutic interventions aimed at correcting their transcriptional output.

Functional enhancers are located in both gene-body and distal regions, with 28.2% of them lacking active chromatin markers commonly used for annotating candidate enhancers. Similar findings were reported for enhancers identified from CRISPR screens in mouse embryonic stem cells (mESCs)[22] and H1 human embryonic stem cells (hESCs)[15], and transgenic mouse reporter assays[23]. These findings reinforce the concept of the existence of hidden enhancers that do not

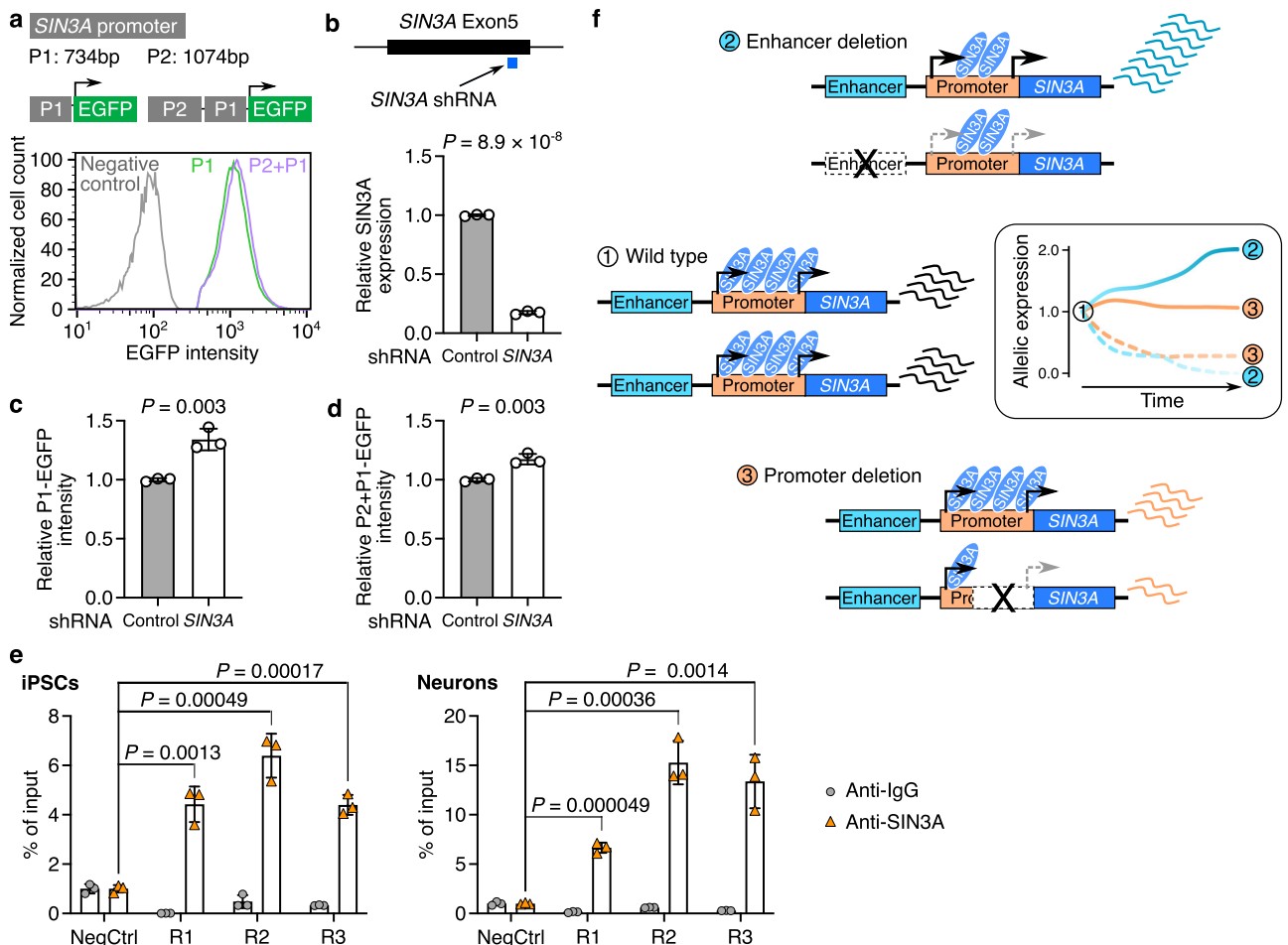

**Fig. 5 | The *SIN3A* promoter mediates allelic enhancer deletion-induced allelic compensation effect (ACE). a** Flow cytometry plots showing the EGFP expression from *SIN3A* promoter reporters. **b** shRNA-mediated downregulation of *SIN3A*. **c**, **d** *SIN3A* promoter reporters show significantly higher EGFP intensity in cells with *SIN3A* shRNA, compared to cells with control shRNA. **e** SIN3A binding at the *SIN3A* promoter region assessed by ChIP-qPCR. R1: Region 1; R2: Region 2; R3: Region 3. Data in (**b**–**e**) are from three biological replicates and are shown as mean ± SD. *P*

values in (**b**–**e**) were determined using the two-tailed two-sample *t*-test. **f** The working model of allelic enhancer deletion-induced ACE. *SIN3A* is evenly expressed from two alleles in wild-type cells. Allelic enhancer deletion causes downregulation of *SIN3A* from the enhancer deletion allele (sky blue dashed line), which triggers ACE from the intact allele (sky blue solid line). Allelic partial promoter deletion causes partial downregulation of *SIN3A* (orange dashed line) without ACE (orange solid line). Source data are provided as a Source data file.

have typical epigenetic features and emphasize the importance of characterizing regulatory elements in an unbiased manner with functional assays. In addition, using dense sgRNA pairs improves the likelihood of identifying functional enhancers, considering the varied efficiency of sgRNAs. We observed that only 41% of CREST-seq identified enhancers physically interact with their target gene promoters in neurons, which could be attributed to two possibilities. One is that mechanisms other than chromatin interactions, including RNA polymerase tracking, TFs linking, and enhancer relocation, are used for transcriptional regulation[42]. Another possibility is that our study can identify enhancers that contribute to gene expression during the differentiation process, and their interaction with the promoter occurs in cells before they differentiate into neurons.

At the individual pgRNA level and DNA fragment level, we observed some variation between replicates. We suspect this variation stems from the low number of cells collected from the sorted samples (approximately the bottom 1% of cells with reduced expression), which may have led to the exclusion of weaker enhancers in our screens. We advise increasing the number of collected cells and incorporating additional replicates when feasible to enhance the robustness of results when using the CREST-seq method.

Enhancers outnumber protein-coding promoters, highlighting the complexity of the gene regulatory program, which remains inadequately comprehended. The "enhancer" terminology encompasses a variety of different classes of enhancers with distinct functional consequences on gene regulation. For example, some enhancers are redundant and may only cause transient transcriptional disruption when deleted[43,44]. The redundancy within the enhancer program is advantageous for achieving precise and resilient gene expression. Other enhancers, such as shadow enhancers[45], exert an additive function in transcriptional output, whereby multiple enhancers collectively contribute to the desired transcriptional level of target genes[46,47]. Our study unveils an additional layer of complexity to the gene regulation program by uncovering ACE upon allelic enhancer deletion for dosage-sensitive genes.

It is crucial for diploid organisms to maintain finely tuned expression levels for dosage-sensitive genes, including TFs and haploinsufficient genes. These genes play pivotal roles in many fundamental biological processes, and any change in the transcription level of these genes or a loss-of-function mutation on one of the alleles will render them insufficient for their function. Therefore, a precise and robust transcriptional control mechanism must be established to

guarantee optimal transcriptional output. Typically, multiple enhancers participate in regulating a target gene, increasing the vulnerable genomic space subjected to deleterious mutations that could adversely affect transcriptional control. We suggest that ACE is one type of genetic compensation mechanism[48,49], which serves as a defense mechanism for overcoming adverse effects caused by enhancer mutations and accounting for widespread sensitivity to TF dosages during development.

Negative feedback loop (NFL) is a regulatory mechanism that maintains optimized gene dosage across various regulatory layers. At the transcriptional level, for instance, the long noncoding RNA *Chaserr* forms a feedback loop that buffers CHD2 dosage by regulating *Chd2* expression[50]. At the post-transcriptional level, genes autoregulate through a feedback mechanism involving alternative splicing[51] and nonsense-mediated mRNA decay, as seen in *SFPQ*[52], *RPS3*[52], and the serine arginine-rich protein family *SRSF1, SRSF2, SRSF3, SRSF4, SRSF5,* or *SRSF7*[53]. At the post-translational level, a classic example is the p53–MDM2 loop, in which p53 activates *MDM2* expression, and MDM2 in turn promotes p53 degradation[54,55]. Compared to these well-characterized NFLs, ACE represents a distinct form of NFL triggered specifically by allelic *cis*-acting enhancer deletion. ACE extends the framework of NFLs by exemplifying a dosage compensation mechanism that is initiated by enhancer loss and mediated through promoter sensing and transcriptional upregulation.

Our results demonstrate that promoter sequences play a critical role in detecting reduced gene dosage and initiating transcriptional compensation through the binding of their own protein products. Compared to allelic enhancer deletion, we didn't observe transcriptional compensation in the allelic deletion of the *SIN3A* promoter (Figs. 2g and 4b, c, Supplementary Fig. 13c). This is possibly due to the cells with allelic promoter deletion still having sufficient SIN3A binding to the promoter on the other allele. However, the reduced SIN3A level can be directly detected in the cells with allelic enhancer deletion, as they possess two copies of the *SIN3A* promoter. The time course analysis suggests that ACE is more potent as *SIN3A* expression approaches the level that triggers haploinsufficiency. This observation implies that, in addition to ACE, survival selection might contribute to the enrichment of higher SIN3A-expressing cells over time. Our transcriptional compensation model offers one explanation of why disturbing enhancers for dosage-sensitive genes don't seem to affect the cellular and developmental processes. To validate the transcriptional compensation of additional candidate genes, identifying their enhancers and performing allelic gene expression analysis in cells with deletion or perturbation of one copy of enhancer is needed. Further testing of the enhancer-deletion-triggered transcriptional compensation mechanism in vivo will solidify our understanding of how dosage-sensitive genes achieve robust transcriptional output and normal development.

## Methods

### Generating the reporter iPSC lines

To monitor allelic gene expression, we generated C-terminal allelically tagged human iPSC lines for *APP* (*APP-EGFP/mCherry*), *SIN3A* (*SIN3A-EGFP/mCherry*), *FMR1* (*FMR1-mCherry*), and *MECP2* (*MECP2-EGFP*) using CRISPR/Cas9-mediated homology-directed repair (HDR). The parental cell line we used was the WTC11 i³N iPSC line, which has doxycycline-inducible *Ngn2* integrated at the AAVS1 safe harbor locus. The WTC11 i³N iPSC line was a gift from Li Gan's lab. For *SIN3A* and *APP*, we generated EGFP and mCherry donor vectors with identical homology arms. For *MECP2* and *FMR1*, we generated an EGFP donor for *MECP2* and a mCherry donor for *FMR1*. We designed sgRNAs with the targeting site within 100 bp upstream or downstream of each stop codon to knock in the reporters at the C-terminus of the coding region of each gene. We amplified the genomic regions of 500–1000 bp upstream and downstream of the stop codon for each target gene as homology arms and inserted the EGFP or mCherry sequences between homology arms. To prevent EGFP and mCherry from affecting target gene function, we added a GS linker and T2A sequence between the C-terminal of the target gene and the N-terminal of EGFP or mCherry. We also mutated sgRNA target sites or PAM sequences on donor vectors to prevent the CRISPR/Cas9 system from cutting donor vectors during the HDR, without altering the encoded amino acids. We cloned all donor vectors by Gibson assembly (NEB, E2621S) and verified them through Sanger sequencing.

We in vitro transcribed all sgRNA using the Precision gRNA Synthesis Kit (Invitrogen, A29377), and obtained Cas9-NLS protein from QB3 MacroLab at the University of California, Berkeley. We delivered the CRISPR/Cas9 machinery into iPSC in ribonucleoprotein (RNP) format and donor vectors in plasmid format. To assemble RNP complex, we incubated the in vitro transcribed sgRNAs with Cas9-NLS protein at 20-25 °C for 15 min. We then mixed the assembled RNP complex with EGFP and/or mCherry donor vectors and delivered them into WTC11 i³N iPSCs using nucleofection (Lonza, VPH-5012). After nucleofection, we seeded the cells into Matrigel-coated (Corning, 354277) wells for recovery. Three to four days later, we sorted the EGFP and mCherry double-positive cells (for *SIN3A* and *APP*), EGFP-positive cells (for *MECP2*), or mCherry-positive cells (for *FMR1*) into Matrigel-coated (Corning, 354277) 96-well plates with one cell per well using fluorescence-activated cell sorting (FACS) to generate clonal allelically tagged reporter cell lines. After about 2 weeks, we expanded the viable clones and analyzed them to establish reporter cell lines. We validated the individual clonal reporter cell lines with genotyping PCR, followed by Sanger sequencing and flow cytometry analysis. The step-by-step protocol can be found at STAR Protocols[56]. DNA sequences of oligos and sgRNAs are listed in Supplementary Data 2 and 3.

### Cell culture and neuronal differentiation

The WTC11 i³N iPSCs were cultured on Matrigel-coated (Corning, 354277) plates and maintained in Essential 8 medium (Thermo Fisher Scientific, A1517001), and passaged with Accutase (STEMCELL Technologies, 07920) and 10-µM ROCK inhibitor Y-27632 (STEMCELL Technologies, 72302). The Human embryonic kidney (HEK) 293T cells were cultured in Dulbecco's Modified Eagle medium (Gibco, 11995065) with 10% fetal bovine serum (FBS) (HyClone, SH30396.03), and passaged with trypsin-EDTA (Gibco, 25200072). All the cells were grown with 5% $CO_2$ at 37 °C and verified mycoplasma-free using the MycoAlert Mycoplasma Detection Kit (Lonza, LT07-218). The differentiation of WTC11 i³N iPSCs into excitatory neurons was performed using a two-step differentiation protocol. Briefly, iPSCs were cultured on Matrigel-coated plates with pre-differentiation media containing doxycycline (2 µg/mL; Sigma-Aldrich, D9891) for 3 days, with daily media changes. After 3 days, the pre-differentiated cells were dissociated with Accutase (STEMCELL Technologies, 07920) and replated onto Poly-L-Ornithine-coated (15 µg/mL; Sigma-Aldrich, P3655) plates with maturation media containing doxycycline. The maturation media were changed weekly by removing half of the media from each well and adding an equal amount of fresh media without doxycycline. The detailed protocol is accessible at the ENCODE portal (https://www.encodeproject.org/documents/d74fb151-366c-4450-9fa0-31cc614035f9/).

### sgRNA library design, cloning, packaging

To perform tiling deletion CRISPR screens, we designed paired sgRNA (pgRNA) library for each target locus, including *SIN3A* (chr15: 74,370,000-76,461,000, hg38), *APP* (chr21: 24,880,000-27,180,000, hg38), *FMR1* (chrX: 146,000,000-150,000,000, hg38), and *MECP2* (chrX: 153,000,000-155,100,000). We first selected all the available sgRNAs within each target region from the sgRNA database generated in CREST-seq[15] and added a G at the start of the sgRNAs that didn't start with G. Then, we removed sgRNAs containing any transcriptional

termination sequences (AATAAA, TTTTT, TTTTTT) or BsmBI cut sites (CGTCTC, GAGACG). After filtering, we paired sgRNAs sequentially to generate pgRNA libraries. For each library, the average distance between each sgRNA pair is about 2000–3000 bp, and the average coverage of sgRNA pairs across each nucleotide in the target region is 15 or 20. To design non-targeting negative control pgRNAs, we first identified unique 20-bp-long DNA sequences that weren't followed by the NGG PAM sequence and added a G at the start of the sequences that didn't start with G. We removed DNA sequences containing any sequence of TTT, TTNTT, TTTTTT, AATAAA, AAAAA, CGTCTC, or GAGACG. Next, we paired them into pairs with an average distance between two sequences about 1500–2000 bp. For positive control pgRNAs targeting EGFP and mCherry, we manually designed 10 sgRNAs targeting EGFP or mCherry sequence and named them with numbers 1 to 10 according to their locations in EGFP or mCherry sequence from N-terminal to C-terminal. We further generated pgRNAs by pairing sgRNA1 to sgRNA6, sgRNA2 to sgRNA7, and so on. The oligo libraries of *APP*, *FMR1*, and *SIN3A* were made by following the template of CTTGGAGAAAAGCCTTGTTT{sgRNA1}GTTTAGA-GACG{10nt_random_sequence}CGTCTCACACC{sgRNA2}GTTTTA-GAGCTAGAAATAGCAAGTT, and the oligo library of *MECP2* was made by following the template of TGTGGAAAGGACGAAACACC{sgRNA1}GTTTAAGAGACG{10nt_random_sequence}CGTCTCTTGTTT{sgRNA2}GTTTTAGAGCTAGAAATAGCAAGTT. We synthesized the designed pgRNA libraries (Twist Bioscience) and cloned into lentiCRISPRv2 plasmid with mouse U6 promoter for *APP*, *FMR1* and *SIN3A*, and cloned into lentiCRISPRv2 plasmid with human U6 promoter for *MECP2*.

We used a two-step cloning strategy to clone these pgRNA libraries. First, we amplified the pgRNAs from the synthesized oligo pool with NEBNext High-Fidelity 2× PCR Master Mix (NEB, M0541S). For each 50 µl PCR reaction, we used 0.5 µl 20 nM oligo pool as a template. The PCR reaction was performed as follows: 98 °C 30 s; 98 °C 10 s, 55 °C 30 s, 72 °C 30 s, for 15 cycles; 72 °C 5 min; 4 °C hold. The amplified oligo pool was purified and inserted into BsmBI-digested lentiCRISPRv2 plasmids via Gibson assembly (NEB, E2621L). The assembled products were transformed into NEB 5-α electrocompetent *Escherichia coli* cells (NEB, C2989K) by electroporation. Millions (1000× of pgRNA library size) of independent bacterial colonies were cultured, and pgRNA library plasmids from first-step cloning were extracted with the Qiagen EndoFree Plasmid Mega Kit (Qiagen, 12381). Second, we digested the pgRNA library plasmids from first-step cloning with BsmBI and purified the product with gel extraction (MACHEREY-NAGEL, 740609.250S). Then, a DNA fragment containing a sgRNA scaffold and another U6 promoter was ligated to the BsmBI-digested pgRNA library plasmids using T4 ligase (NEB, M0202M). The ligated products were electroporated into NEB 5-α electrocompetent *Escherichia coli* cells (NEB, C2989K), and millions (1000× of pgRNA library size) of bacterial colonies were cultured for each library. The final plasmid libraries were extracted with the Qiagen EndoFree Plasmid Mega Kit (QIAGEN, 12381). To check the quality of each pgRNA plasmid library, we amplified the pgRNA cassette from the cloned plasmid library by three rounds of PCR with NEBNext High-Fidelity 2× PCR Master Mix (NEB, M0541S). The DNA sequences of oligos used for pgRNA library cloning are listed in Supplementary Data 2. The prepared libraries were sequenced with paired-end deep sequencing.

Four pgRNA libraries were packaged into lentivirus libraries individually in HEK293T cells. The titration of each lentivirus library was tested in its associated reporter iPSC lines. The detailed steps for lentivirus packaging and titration were the same as previously described[57]. To make the lentiviral library, 5 µg of pgRNA plasmid library was cotransfected with 3 µg of psPAX (Addgene, 12260) and 1 µg of pMD2.G (Addgene, 12259) lentivirus packaging plasmids into 8 million HEK293T cells in a 10-cm dish with PolyJet (SignaGen Laboratories, SL100688). The medium was replaced 12 h after transfection and harvested every 24 h for a total of three harvests. Harvested media

containing the desired virus were filtered through Millex-HV 0.45-µm polyvinylidene difluoride filters (Millipore, SLHV033RS) and further concentrated with 100,000 NMWL (nominal molecular weight limit) Ultra-15 centrifugal filter units (Amicon, UFC910008). The titer of lentivirus was determined by transducing 500,000 cells with varying amounts (0, 0.5, 1.0, 2.0, 4.0, and 8.0 µl) of concentrated virus and polybrene (8 µg/ml; Millipore, TR-1003-G). Viral transduction was performed by centrifuging the lentivirus and cell combination at 1000 relative centrifugal force (RCF) for 90 min at 37 °C. Three to 4 h later, virus-containing medium was replaced with fresh medium. Twenty-four hours after the transduction, transduced cells were dissociated with Accutase (STEMCELL Technologies, 07920) and seeded as duplicates. One replicate was treated with puromycin (500 ng/mL; Sigma-Aldrich, P8833), and the other replicate was not treated with puromycin. Four days later, the Puromycin-resistant cells and control cells were counted to calculate the ratio of infected cells and the viral titer.

## CREST-seq screen

To identify enhancers for *APP*, *FMR1*, *MECP2*, and *SIN3A*, we performed CREST-seq screens in excitatory neurons differentiated from each reporter cell line, with two biological replicates per gene. For each screen, we seeded the reporter iPSCs in Matrigel-coated 6-well plates with one million cells per well, and the total cell number was about 2,000 times the total oligo number in each pgRNA library. 24 h later, we transduced the lentiviral library into the iPSCs at a multiplicity of infection (MOI) of 0.5 with polybrene (8 µg/mL; Millipore, TR-1003-G) and spun at 1000 RCF at 37 °C for 90 min. The next day, we passaged the infected cells with Accutase (STEMCELL Technologies, 07920) and treated them with puromycin (500 ng/mL; Sigma-Aldrich, P8833) at 48 h after infection for 7 days to get rid of uninfected cells. Then, we differentiated the infected cells into excitatory neurons. Two weeks after differentiation, we treated the excitatory neurons with Papain (20 U/mL; Sigma, P4762) and DNase I (100 U/mL; Sigma, DN25) for 30 min at 37 °C to dissociate them into single cells. We collected the dissociated neurons with DMEM/F12 media (Gibco, 11330032) plus 10% FBS (HyClone, SH30396.03) and pelleted at 200 RCF and 25 °C for 10 min. We resuspended the cell pellets in the HBSS buffer (Gibco, 14175095) with 0.5% FBS for FACS. We collected about 500,000 cells with reduced expression of EGFP or mCherry reporter for each screen. We extracted the genomic DNA from FACS-isolated cells and control cells without FACS via cell lysis and digestion (100 mM, pH 8.5 Tris-HCl, 5 mM EDTA, 200 mM NaCl, 0.2% SDS, and 100 µg/mL proteinase K), phenol: chloroform (Thermo Fisher Scientific, 17908) extraction, and isopropanol (Fisher Scientific, BP2618500) precipitation. We amplified the pgRNA cassette from the genomic DNA by performing three rounds of PCR using 500 ng of genomic DNA for each reaction and NEBNext High-Fidelity 2× PCR Master Mix (NEB, M0541S). We deep sequenced the purified libraries with paired-end sequencing. Detailed information on screening is available at the ENCODE portal (https://www.encodeproject.org/documents/c1194c4c-ba28-4e37-a13f-3dde86d03241/). The DNA sequences of oligos used for pgRNA libraries preparation are listed in Supplementary Data 2.

## Analysis of CREST-seq screens

To quantify the frequency of pgRNAs in each sample, we aligned the paired-end sequencing data to the sequences of designed pgRNAs using BWA (bwa-0.7.17)[58] with default parameters, and only the paired reads that exactly matched the designed pgRNA were counted as the frequency of each pgRNA. To evaluate the performance of the FACS-based screening strategy we used for CREST-seq screens, we checked the fold change and *P* value of each pgRNA in each screen by comparing libraries made from sorted cells and control libraries made from unsorted cells. We performed analysis using CRISPY with default settings, and the total mapped reads normalized read counts of each

screen were used as input for CRISPY. For *SIN3A* and *APP* screens, we analyzed the libraries for the EGFP allele and mCherry allele separately. Significant enrichment of positive control pgRNAs targeting EGFP and mCherry demonstrated the success of these screens. We further identified functional enhancers for each target gene using RELICS (v.2.0)[19]. RELICS splits the region of interest into segments and applies a Bayesian hierarchical model to identify functional sequences supported by the screening data. We prepared the input files to provide genomic coordinates and the total mapped reads, normalized read counts of each pgRNA in the standard input format for RELICS. We labeled pgRNAs overlapping 5′TUR and exons of each target gene as known functional sequences and the designed negative controls as negative controls for RELICS. Then, RELICS identified the functional sequences for each screen using the default settings for RELICS v.2.0 (min_FS_nr:30, glmm_negativeTraining:negative_control, crisprSystem:dualCRISPR). We merged the identified adjacent functional sequences and calculated the median RELICS score for each merged DNA fragment using bedtools (v2.26.0). The merged fragments with a median RELICS score >0.2 and at least two functional sequences were considered enhancers.

## Chromatin signature analysis of identified enhancers

We checked the overlap between chromatin signatures and identified enhancers using bedtools intersect (v2.26.0). For the marks including H3K4me1, H3K4me3, H3K27ac, H3K36me3, H3K9me3, CTCF, and RNA polymerase II, we downloaded the original sequencing files from Gene Expression Omnibus database under accession number GSE167259. We aligned them to the GRCh38/hg38 reference genome using ENCODE chip-seq-pipeline2 (v2.1.6) with the standard setting. We used the overlap optimal peaks for chromatin signature analysis. For cCREs in excitatory neurons from the human brain samples, we downloaded the bed files containing identified cCREs from 38 excitatory neuron subtypes[21] (www.catlas.org) and merged them together using bedtools (v2.26.0). We used the merged bed file containing all the cCREs in excitatory neurons for chromatin signature analysis. For accessible genomic regions, we used the ATAC-seq peaks identified in WTC11 i³N iPSC-derived excitatory neurons[17].

## Validation of identified enhancers

We performed the validation experiments for enhancers and promoters by using paired sgRNA-mediated CRISPR deletion. For each region, we designed two sgRNAs to delete the target region (sgRNA sequences are listed in Supplementary Data 3). To clone the two sgRNAs into lentiCRISPRv2 vector (Addgene, #52961), we amplified the sgRNA scaffold and mouse U6 promoter using two oligos containing the designed sgRNA sequences, and inserted the amplified DNA fragments into the lentiCRISPR v2 vector (Addgene, #52961) using Gibson assembly (NEB, E2621L). The resulting plasmid contains two sgRNAs with the pattern of hU6-sgRNA1-mU6-sgRNA2. After validating the sgRNA sequences via Sanger sequencing, we individually packaged each plasmid into lentivirus using the same procedure used to package the pgRNA lentiviral library. We performed validation experiments individually by infecting the reporter cell lines with the associated lentivirus. About 200,000 reporter iPSCs were seeded into a Matrigel-coated cell in a 24-well plate, and the cells were infected with lentivirus 24 h after seeding using the spin infection method we used for the CREST-seq screen. Forty-eight hours after infection, we treated the cells with puromycin (500 ng/mL; Sigma-Aldrich, P8833) for 7 days. For the validation in iPSCs, we cultured the infected iPSCs for 7 days without puromycin treatment and performed flow cytometry analysis. For the validation in excitatory neurons, we differentiated the infected iPSCs into excitatory neurons and analyzed the neurons with flow cytometry at 14 days after differentiation. For *SIN3A* and *MECP2* validations, we established single-cell clones using FACS-mediated single-cell sorting. We sorted the single cells with reduced

expression of the reporter into 96-well plates with one cell per well. About 2 weeks later, we transferred the viable clones to 24-well plates to establish single-cell clones and randomly choose two clones for each analysis.

## Flow cytometry analysis and fluorescence-activated cell sorting

The cells for flow cytometry analysis and fluorescence-activated cell sorting (FACS) were dissociated into single cells using Accutase (STEMCELL Technologies, 07920) for iPSCs and Papain (Sigma, P4762) for excitatory neurons. The iPSCs were resuspended with FACS buffer (1× DPBS, 2 mM EDTA, 25 mM HEPES pH 7.0, and 1% FBS), and neurons were resuspended with HBSS buffer (Gibco, 14175095) with 0.5% FBS. We used the same gate setting for both flow cytometry analysis and FACS. First, cells were separated from the debris based on the forward scatter area (FSC-A) and side scatter area (SSC-A). Then, single cells were separated using a single cell gate based on the width and area metrics of the forward scatter (FSC-W versus FSC-A) and side scatter (SSC-W versus SSC-A). Further, the gates for EGFP and mCherry signal baselines were set using cells without EGFP and mCherry signals. Flow cytometry analyses were performed on BD LSR II and BD LSRFortessa Flow Cytometers. FACS experiments were conducted on a BD FACSAria II instrument using a 100-μm nozzle. All the plots associated with flow cytometry analysis and FACS were made using FlowJo (v10.7.2).

## Time-course analysis of *SIN3A* transcriptional compensation

To monitor the transcriptional compensation of *SIN3A*, we seeded *SIN3A-EGFP/mCherry* iPSCs in a Matrigel-coated 12-well plate with 200,000 cells per well. 24 h later, we infected the cells with lentivirus expressing Cas9 and pgRNAs targeting the *SIN3A* promoter and an enhancer. After infection, we dissociated the cells with Accutase (STEMCELL Technologies, 07920) at each time point. We used one-third of the cells for flow cytometry analysis and maintained two-thirds for analysis at the next time point. We analyzed the cells using BD LSRFortessa Flow Cytometers and analyzed the data using FlowJo (v10.7.2).

## RT-qPCR and allelic gene expression

We extracted total RNA from each sample using QIAGEN plus mini RNA kit (Qiagen, 74134), and 1 μg total RNA was used to make cDNA with iScript cDNA synthesis kit (Bio-Rad, 1708891). To check the allelic expression of *SIN3A*, we used one SNP located in the *SIN3A* intron. We amplified the SNP region from each cDNA sample and added deep sequencing adapters via PCR to prepare a sequencing library for each sample. The amplicons in each purified library were analyzed by deep sequencing (DNA oligos are listed in Supplementary Data 2). The copy number of each allele of *SIN3A* in each sample was counted using a 21 bp window with the SNP in the middle. The total expression levels of *SIN3A* and *MECP2* were analyzed on a Roche LightCycler 96 instrument using Luminaris HiGreen qPCR Master Mix (Thermo Scientific, K0992) (DNA oligos are listed in Supplementary Data 2). Data were normalized to *GAPDH*.

## CTCF motif deletion

We scanned transcription factor motifs in SIN3A-E4 using FIMO (v5.4.1)[59] with human motif database HOCOMOCO (HOCOMOCOv11 full annotation)[34] and default settings. We focused on a CTCF motif and designed two sgRNAs with spacer sequences overlapping CTCF motifs. We cloned the two sgRNAs into the lentiCRISPRv2 vector (Addgene, #52961) individually and packaged them into lentivirus. We infected the SIN3A-EGFP/mCherry iPSCs with each lentivirus separately and treated the cells with puromycin for 7 days. After puromycin treatment, we cultured the cells for an additional 7 days. We then isolated cells with reduced expression levels of EGFP or mCherry reporters from each cell pool using FACS and extracted the genomic DNA from these isolated cells. To check the DNA sequences in the sgRNA

targeting sites, we amplified the sgRNA target sites with PCR and deep sequenced the amplicons (DNA oligos are listed in Supplementary Data 2). The deep sequencing data of each sample were analyzed using CRISPRssor2[60]. Topologically associating domain (TAD) information at the SIN3A-E4 locus was obtained from WTC11 iPSC Micro-C data (https://data.4dnucleome.org/, experiment ID: 4DNESODGV2V2).

### The overexpression of *SIN3A*
To overexpress *SIN3A*, we constructed the *SIN3A* promoter P1-controlled *SIN3A* expression plasmid (SIN3A-Pr>SIN3A-P2A-BFP) and CAG promoter-controlled *SIN3A* expression plasmid (CAG>SIN3A-P2A-BFP). We amplified the *SIN3A* promoter P1 region (chr15: 75451566 - 75452299, hg38) from the genomic DNA of WTC11 i³N iPSCs, CAG promoter from a plasmid (Addgene, #220439), SIN3A coding region from cDNA made from total mRNA of WTC11 i³N iPSCs, and BFP from a plasmid (Addgene, #102244) using NEBNext High-Fidelity 2× PCR Master Mix (NEB, M0541S). We inserted three fragments, including the promoter (SIN3A or CAG promoter), SIN3A coding region, and BFP, into the pLS-SceI plasmid (Addgene, #137725) and replaced the minimal promoter and EGFP sequences using Gibson assembly (NEB, E2621L) to construct the *SIN3A* expression plasmids. We packaged the *SIN3A* expression plasmids into lentivirus and delivered them into iPSCs via spin infection. To check the expression level of *SIN3A* in the infected cells, we isolated BFP-positive cells using FACS and extracted the total mRNA from BFP-positive cells using QIAGEN plus mini RNA kit (Qiagen, 74134), and 1 μg total RNA was used to make cDNA with iScript cDNA synthesis kit (Bio-Rad, 1708891). The total *SIN3A* expression levels were analyzed on a Roche LightCycler 96 instrument using Luminaris HiGreen qPCR Master Mix (Thermo Scientific, K0992). The DNA sequences of oligos are listed in Supplementary Data 2. Data were normalized to *GAPDH*.

### *SIN3A* promoter reporter assay
We used *SIN3A* promoter reporter to test the transcriptional activity of the *SIN3A* promoter under wild-type and SIN3A knockdown conditions. To construct the *SIN3A* promoter reporter plasmids, we modified one lentivirus EGFP reporter plasmid (Addgene #137725) by replacing the scaffold-attached region (SAR)[61,62] with human anti-repressor element 40[63], and used the modified plasmid as a backbone for *SIN3A* promoter reporter plasmid cloning.

We picked two regions (P1, Chr15: 75451566-75452299; P2, Chr15: 75453777-75454850) as *SIN3A* promoter based on the ATAC-seq and SIN3A ChIP-seq data. We amplified these two regions from the genomic DNA of WTC11 i³N iPSCs and inserted them before the start of the EGFP sequence in the modified EGFP plasmid via Gibson assembly (NEB, E2621L), and constructed P1-EGFP and P1 + P2-EGFP reporter plasmids. To knockdown *SIN3A* expression, we used shRNA-mediated knockdown. We designed a shRNA targeting *SIN3A* mRNA using the DSIR tool (http://biodev.extra.cea.fr/DSIR/DSIR.html) and used a human control shRNA from a previous study[64]. To clone shRNA expression plasmids, we replaced the sgRNA scaffold and Cas9 expression cassette in lentiCRISPRv2 (Addgene, #52961) vector with EF1α-HygR-BFP. shRNAs were cloned into the modified lentiCRISPRv2 vector under the control of a human U6 promoter and packaged into lentivirus for cell transduction. The cloned plasmids were verified using Sanger sequencing and packaged into lentivirus. To test the knockdown efficiency of *SIN3A* shRNA, we infected SIN3A-EGFP/mCherry reporter cell line with lentivirus containing *SIN3A* shRNA or control shRNA and checked the SIN3A-EGFP and SIN3A-mCherry signals with flow cytometry 6 days after infection. We used the EGFP and mCherry signals from WTC11 i³N cells as baselines and calculated the knockdown efficiency of *SIN3A* shRNA relative to control shRNA. The average knockdown efficiency from SIN3A-EGFP and SIN3A-mChery alleles was used as knockdown efficiency of *SIN3A* shRNA. To test the *SIN3A* promoter reporter, we infected WTC11 i³N iPSCs with P1-EGFP

and P1 + P2-EGFP lentivirus individually with MOI < 0.1, and isolated the EGFP-positive cells using FACS. Then, we infected the FACS-isolated P1-EGFP and P1 + P2-EGFP cells with lentivirus containing control shRNA and *SIN3A* shRNA individually and checked the EGFP signal with flow cytometry 6 days after infection. We performed all the experiments in three biological replicates and analyzed them with BD LSRFortessa Flow Cytometer and FlowJo (v10.7.2). The sequences of shRNAs are listed in Supplementary Data 4.

### Identification and analysis of candidate transcriptional compensation genes
To identify candidate transcriptional compensation genes, we extracted the promoter sequences (±1 kb of TSS) for each protein-coding gene in the human (GENCODE v44 annotation) and mouse (GENCODE vm33 annotation) genomes. Then, we searched for transcription factor (TF) binding motifs in these promoter sequences using FIMO (v5.5.4)[59] (P < 0.0001) and TF motifs from HOCOMOCO (HOCOMOCOv11 full annotation)[34] and JASPER (JASPAR2022 CORE vertebrates)[33] databases. GO term analysis was performed using Enrichr[65]. The identity of each TF was annotated using UniProtKB (activator or repressor) and Gene Ontology (AmiGO 2 with the terms "DNA-binding transcription activator activity" or "DNA-binding transcription repressor activity"). The expression of the identified candidate transcriptional compensation genes was checked using bulk tissue RNA-seq data from GTEx (RNASeQCv1.1.9)[35].

### Allelic analysis of *SIN3A* enhancer-mediated *cis*-regulation
We identified phased SNPs using WTC11 whole genome sequence data (https://www.allencell.org/genomics.html). To perform allelic analysis of *SIN3A* enhancer-mediated cis-regulation, we selected one phased SNP located in the last intron of *SIN3A* (chr15: 75374632, C/T, hg38) and another phased SNP near SIN3A-E2 (chr15: 74721849, T/G, hg38). To link the *SIN3A* alleles to the tagged EGFP and mCherry reporters, we amplified the genomic region covering the *SIN3A* intron SNP and reporters using TaKaRa LA Taq DNA Polymerase (TaKaRa, RR042A), genomic DNA from SIN3A-EGFP/mCherry iPSCs, and reporter-specific primers (GFP-Rs1, mCherry-Rs1, SIN3A_intron_SNP1-R). Then, we sequenced the PCR product using Sanger sequencing to confirm the relationship between *SIN3A* intron SNP and reporters. To check the enhancer deletion allele, we infected the SIN3A-EGFP/mCherry iPSCs with lentivirus expressing Cas9 and pgRNAs targeting SIN3A-E2, followed by puromycin treatment for 7 days. Then, we isolated the cells with reduced expression levels of *SIN3A-EGFP* or *SIN3A-mCherry* using FACS. We extracted the genomic DNA from FACS-isolated cells using QuickExtract DNA Extraction Solution (Biosearch Technologies, QE0905T). We amplified the allele with enhancer deletion from each genomic DNA using TaKaRa LA Taq DNA Polymerase (TaKaRa, RR042A) and primers targeting the SIN3A-E2 region (SIN3A_En_SNP-F, SIN3A_En_SNP-R). We then performed TOPO cloning (Invitrogen, 450071) and sequenced 6 colonies from each sample using Sanger sequencing to verify the sequences. The DNA sequences of oligos used in this experiment are listed in Supplementary Data 2.

### ChIP-qPCR
Chromatin immunoprecipitation followed by quantitative PCR (ChIP-qPCR) was performed as described[66] with minor modifications. WTC11 i3N iPSCs were detached with Accutase (STEMCELL Technologies, 07920), collected with DPBS (Gibco, 14190144), and centrifuged at 200 × g for 10 min at room temperature. The collected WTC11 cells were cross-linked in 1% formaldehyde for 10 min at room temperature, followed by quenching with 125 mM glycine for 5 min. The differentiated 2-week neurons were rinsed twice with HBSS buffer (Gibco, 14175095), and cross-linked in 1% formaldehyde in the culture dish for 10 min at room temperature, followed by quenching with 125 mM glycine for 5 min. The cross-linked neurons were collected with cell

scraper. Then, cross-linked cells were washed twice with ice-cold DPBS (Gibco, 14190144) and centrifuged at $1000 \times g$ for 10 min at 4 °C. The resulting cross-linked cells were incubated with lysis buffer (20 mM Tris·HCl, pH 8.0, 0.5% SDS, 2 mM EDTA, 150 mM NaCl, 1% Triton X-100, 1× Protease inhibitor) for 20 min on ice. For each sample, 6 million lysed cells (3 million lysed cells in 130 μL lysis buffer for one micro-TUBE AFA Fiber Pre-Slit Snap-Cap 6x16mm tube) were sonicated using Covaris S220 focused-ultrasonicator (Duty factor, 5%; Peak incident power, 105 W; Cycles per burst, 200) for 10 min 30 s to shear chromatin into 200–500 bp fragments. The sonicated lysate was centrifuged at $12,000 \times g$ for 10 min at 4 °C, and 20 μL of the sheared chromatin was saved as input. The remaining sheared chromatin was evenly split for incubating either with anti-IgG (Antibodies-Online, ABIN101961) or anti-SIN3A (Novus Biologicals, NB600-1263). After precleared with 30 μL Dynabeads Protein A beads (Invitrogen, 10001D) and 760 μL dilution buffer (10 mM Tris·HCl pH 8.0, 159 mM NaCl, 1.14 mM EDTA, 1.14% Triton X-100, 0.1% SDS, 0.114% Sodium Deoxycholate, 1× Protease inhibitor) for 1 h at 4 °C on a rotator, the precleared chromatin was further incubated with 2.5 μg anti-IgG or anti-SIN3A at 4 °C overnight on a rotator. Meanwhile, the Protein A beads that will be used in the chromatin immunoprecipitation were blocked with BSA buffer (10 mM Tris·HCl, pH 8.0, 140 mM NaCl, 1 mM EDTA, 1% Triton X-100, 0.1% SDS, 0.1% Sodium Deoxycholate, 5 mg/mL BSA) at 4 °C overnight on a rotator. Next day, the antibody-bound chromatin was incubated with BSA-blocked beads at 4 °C for 4 h on a rotator. All samples were then washed three times with low salt wash buffer (10 mM Tris·HCl pH 8.0, 140 mM NaCl, 1 mM EDTA, 1% Triton X-100, 0.1% SDS, 0.10% Sodium Deoxycholate, 1× Protease Inhibitor Cocktail), two times with high salt wash buffer (10 mM Tris·HCl pH 8.0, 300 mM NaCl, 1 mM EDTA, 1% Triton X-100, 0.1% SDS, 0.10% Sodium Deoxycholate), one time with LiCl buffer (10 mM Tris·HCl pH 8.0, 150 mM Lithium Chloride, 1 mM EDTA, 0.5% IGEPAL CA-630, 0.1% Sodium Deoxycholate), two times with LTE buffer (10 mM Tris·HCl pH 8.0, 0.1 mM EDTA). Subsequently, 200 μL elution buffer (10 mM Tris·HCl, pH 8.0, 1 mM EDTA, 1% SDS) was added to all the samples and inputs and incubated overnight at 65 °C. All the samples were further treated with 4 μg RNase (NEB, T3018L) for 30 min at 37 °C, and 8 μg Proteinase K (NEB, P8107S) for 1 h at 55 °C. Finally, reverse crosslinked samples were purified using AMPure XP beads (Beckman Coulter, A63881) and eluted with 30 μL LTE buffer and analyzed on a Roche LightCycler 96 instrument using Luminaris HiGreen qPCR Master Mix (Thermo Scientific, K0992). ChIP-qPCR was performed with three biological replicates, and data were normalized to the input. The DNA sequences of oligos are listed in Supplementary Data 2.

### ddPCR
The allelic expression of *NANOG* in *NANOG* e1 deletion clones was examined using ddPCR. The original data were provided by Dr. Jielin Yan and Dr. Danwei Huangfu at Memorial Sloan Kettering Cancer Center.

### Ethics statement
The use of the WTC11 iPSCs was approved by the Human Gamete, Embryo and Stem Cell Research Committee at UCSF.

### Reporting summary
Further information on research design is available in the Nature Portfolio Reporting Summary linked to this article.

## Data availability
The CRISPR screen datasets used in this study are available at the ENCODE portal (www.encodeproject.org) and accession numbers are ENCSR783CGW (APP pgRNA plasmid library), ENCSR364KFC (APP control), ENCSR678GDA (Low APP-EGFP), ENCSR952RDF (Low APP-

mCherry), ENCSR493NRD (SIN3A pgRNA plasmid library), ENCSR284PQK (SIN3A control), ENCSR113CEG (Low SIN3A-mCherry), ENCSR750UIY (Low SIN3A-EGFP), ENCSR888FDQ (FMR1 pgRNA plasmid library), ENCSR466IBU (FMR1 control), ENCSR562YXE (Low FMR1-mCherry), ENCSR473BRJ (MECP2 pgRNA plasmid library), ENCSR072YHQ (MECP2 control), and ENCSR119JRG (Low MECP2-EGFP). Public datasets used in this study are listed in Supplementary Data 5. Data can be visualized on the WashU Epigenome Browser using the following session: https://epigenomegateway.wustl.edu/browser2022/?genome=hg38&sessionFile=https://shen-xren.s3.us-west-1.amazonaws.com/CREST-seq_NC/eg-session-YzGAc9x4n-ce92ef10-72ad-11f0-b5d2-a94bbd38dced.json. Tracks include RNA-seq, ATAC-seq, chromatin marks, cCREs annotated in excitatory neurons from the human brain samples, H3K4me3 mediated PLAC-seq, RELICS scores, identified functional enhancers for each screen, and sgRNAs used for validation experiments. Source data are provided with this paper.

## Code availability
Data are analyzed using published pipelines with parameters described in the "Methods" section. No custom code is developed in this study.

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

## Acknowledgements

This work was supported by the NIH grant UM1HG009402 (to Y. Shen and B.R.). This work was made possible in part by the NIH grants P30DK063720 and S101S10OD021822-01 to the UCSF Parnassus Flow Cytometry Core. Sequencing was performed at the UCSF CAT, supported by UCSF PBBR, RRP IMIA, and NIH 1S10OD028511-01 grants. The authors thank Dr. Li Gan at Cornell University for sharing the WTC11 i³N iPSC line. The authors appreciate Dr. Jielin Yan and Dr. Danwei Huangfu at Memorial Sloan Kettering Cancer Center for providing their original data of allelic ddPCR results of *NANOG* expression in *NANOG* e1 deletion clones.

## Author contributions

X.R. and Y. Shen designed the study. X.R. and B.L. designed the pgRNA libraries. X.R., Y.L., L.M., X.C., T.W.T., Y. Sun, J.L., and M.A.T. performed the experiments. X.R., L.Z., Y. Sun, H.L., W.W., and Y. Shen contributed to data analysis and interpretation. X.R. and Y. Shen prepared the manuscript with input from all authors. Y. Shen, W.W., and B.R. supervised the work and obtained funding.

## Competing interests

B.R. is a co-founder and consultant of Arima Genomics Inc. and co-founder of Epigenome Technologies. The other authors declare that they have no competing interests.
