## [Transparent Peer Review file · Nature Communications]

CRISPR tiling deletion screens reveal functional enhancers and allelic compensation effects (ACE) on SIN3A transcription

Corresponding Author: Dr Yin Shen

Version 0:

Reviewer comments:

Reviewer #1

(Remarks to the Author)

Ren et al use a CRISPR based tiling screen to identify functional enhancers of neuropsychiatric risk genes, and obtain evidence of transcriptional allelic compensation effects. This is a very interesting study that I read with great pleasure, and which will deserve publication in a good journal. However prior to that, there are a number of (significant) issues to be addressed:

Comments (in the order of appearance in the manuscript):

-the title indirectly implies to me that functional enhancers of many neuropsychiatric risk genes are assessed; at the end, those are “only” four genes. I think the authors should therefore tone down a bit the title that might give a wrong impression for the reader what is to be expected from the paper. Same applies to the ACE that only is found at the SIN3A locus, and not a general phenomena of the other loci investigated. APP is involved in Alzheimer’s disease. Although Alzheimer’s can have neuropsychiatric symptoms, I am not completely sure that in general Alzheimer is considered a neuropsychiatric disorder? Rather a neurodegenerative one. So the authors should also consider their language use on this point.

-OMIM numbers should be used where in the manuscript known disorders / syndromes are first mentioned (for example in the second paragraph of the introduction).

-around line 74: it should probably be specified that the WTC11 cell used is a male cell line, hence only requiring to tag a single copy of the X-linked genes that are studied.

-around line 80: did the authors check whether there was any specific gRNA depletion at the iPS cell stage, that was no longer present in the induced neurons, which could point to any deletion effects impacting on the differentiation of the induced neurons and might therefore be selected against?

-line 91-104: I appreciate the detailed analysis of the identified enhancers vs various epigenome data sets. It might however further strengthen the data to also compare the identified enhancer to previously established resources for brain related enhancers, to get a glance how many were previously unidentified. For example, the data could be compared to those in the atlas generated in PMID: 34663447 or other related work. Especially for the ones not associated with enhancer associated chromatin marks, it might be interesting to see whether those sequences are actually marked with such marks at other brain developmental stages but just not in the cell type that is being investigated (some kind of priming?)

-line 132-148: I can see the motivation (and appreciate) to show some “clinical utility” of the findings from the screen, by employing CNV enrichment analysis. In its present state however, the analysis is meaningless. Most importantly, if the hypothesis is true (and I don’t have doubts about that) that genetic variants of the functional identified enhancers can contribute to disease, this would certainly be only true for deletions that remove an enhancer, or for other point mutations / small alterations that disrupt the enhancer itself (for example interfering with a TF binding site). Hence, in their CNV analysis, the authors should only focus on deletions (e.g., for CNVs that are duplications, it is by no means clear from usual diagnostic SNP-array data whether such duplications occur in cis / in tandem, or might mess up the 3D organization or even

be integrated in completely different loci) as for all other CNVs it cannot be just assumed that there is a cis-regulatory effect. Also the enrichment analysis from line 140 onwards seems to be completely disconnected from the 39 functional enhancers that the authors identify, now moving to cCREs (identified from where?) identified to 355 disorders? There is already quite some literature that shows enrichment of deletions over potential regulatory elements possibly contributing to disease, so adding this unrelated analysis and concluding with line 147/148 is a bit too much selling of the importance of the findings in my opinion. It would be better if the authors would search for specific deletions overlapping specifically the functional enhancers identified, that could be found in yet undiagnosed patients. There are already great resources available to this (for example the Genomics England 100,000 genome data, data from the GREGOR consortium, and plenty of clinical geneticists likely be excited to collaborate on this as this might help to resolve current "missing heritability"). Also, alternatively, the authors could use publicly available genome data from gnomADv4 to show that their functional enhancers are depleted for variants in the healthy population, further emphasizing their potential role in disease.

-line 150-219: the observation of ACE at the SIN3A locus is highly interesting, and the authors use elegant experiments trying to understand this mechanism. As the authors also mention, my feeling would be that this effect is likely caused at this locus given the essentiality of SIN3A and its autoregulation, which might not tolerate reduced expression. The authors already elegantly check that once the ACE is established in their enhancer deletion clones, overexpression of exogenous SIN3A cannot restore this. I think one experiment that is missing and which should be added would be the reverse. E.g., can the SIN3A exogenous overexpression first be established in the double reporter background, and subsequently the enhancer be deleted. In that case, you would expect that ACE would not occur, given that the SIN3A dosage problem for the cell is solved by the exogenous expression. Adding this experiment could more securely nail down that the observed ACE is due to the essentiality of the gene. In general, if the authors would like to make more broad statements about ACE (which I feel is the case given the manuscript title), my advice would be to chose another essential gene and perform the same analysis (but the simpler solution would be to tone down on the claims and make sure that it is clear that this so far is a locus specific SIN3A observation). Alternatively, the authors should look in literature for other examples and discuss them. To my mind come studies from Pablo Navarro and Ian Chambers looking at the auto-regulation of the NANOG locus in mouse ES cells (PMID: 23178592 and perhaps also later work of them). I am not completely sure as it has been a while that I was working in that field, but I think with reporters at that locus very similar observations had been made when altering one of the upstream NANOG enhancers.

-line 230: the drawn conclusion is wrong: "Both P1 and P1+P2 promoter reporters exhibited a significant increase of promoter activity when endogenous SIN3A expression is reduced by SIN3A shRNA (Fig. 5b-d). Thus, the SIN3A promoter can counteract allelic enhancer deletion-induced downregulation by increasing its transcriptional activity." All what the authors show is that the autoregulation is (as expected) mediated by the SIN3A promoter. The authors simply test promoter activity upon shRNA knockdown of SIN3A in these experiments. That is independent from their earlier findings on the enhancer deletion. The authors should thus rephrase their conclusions accordingly. If the authors want experiments that support their current conclusion, they should look at promoter output in the cells having the SIN3A enhancer deletion and not do the indirect shRNA knockdown/transfection in wild type cells.

-line 244-262: again, this analysis is a bit distractive. I can see why the authors perform this, but once again in the current state the analysis doesn't say much more than that there are plenty of genes encoding for regulatory factors that bind to their own promotor, that are widely expressed, dosage sensitive and therefore like also enriched for being disease relevant genes. However, the analysis does not say for any of these genes, that ACE is happening (e.g., that if the authors would perform the same enhancer deletion for the other loci, the same upregulation of the other allele would happen). Hence I really don't like this effort of overselling the conclusions, and I think the authors should significantly tone down on them. This is not to criticize their experimental findings, as I think the data is very interesting and solid but they should just be described as they are and deliberated from the sauce of overselling in my opinion.

In summary, I really do like the majority of the wet lab experiments performed and are therefore supportive of publications once my concerns are addressed upon revision. Some of the in silico analysis is distractive and not supportive for the case that the authors want to make. The manuscript would benefit from toning down of some of the overselling. The revision in my opinion should focus on 1) the added experiment with first exogenous SIN3A overexpression followed by enhancer deletion to disentangle causality of the observed ACE ; 2) alternative choices to further strengthen the generalizability of the ACE to other loci ; 3) toning down on the slightly overselling of findings to make sure that the conclusions are really supported by the presented data.

Hope this helps, best wishes, Stefan Barakat

Reviewer #2

(Remarks to the Author)

This study identifies noncoding DNA elements that regulate 4 dosage-sensitive and disease-relevant genes in iPSC-derived excitatory neurons. Using 2-guide CRISPR/Cas9 tiled deletion screens, the authors identified 39 cCREs that affect the expression of one of these genes. They dissect the functions of several of these functional elements and discover that deletion of an element that regulates SIN3A leads to a strong decrease in expression from the SIN3A allele in cis and a compensatory in expression from the allele in trans. The authors show that SIN3A, a transcription repressor, directly binds its own promoter, providing a potential molecular mechanism for this phenomenon.

Identification of regulatory elements for dosage-sensitive neuropsychiatric genes is an important and interesting goal.

Discovery of very long-range regulatory elements for SIN3A is supported by nice analysis of heterozygous deletion clones with allele-specific expression readouts. The location of these regulatory elements is surprising (overlapping other genes, located >600Kb away and with a dozen intervening genes). The CRISPR screening tool applied here (CREST-seq) has been applied previously by the authors, but this represents the most extensive application of the tool to my knowledge to date, and has the potential to yield important technical lessons on what this type of genetic deletion screening can find.

Overall, I would support publication if the following major points can be addressed:

Major points:

1. Technical quality and reproducibility of CREST-seq experiments. I am having trouble assessing the overall robustness of the screen and downstream analysis. The validation experiments for many individual elements are very helpful but more information is needed, for example to assess false positive or false negative rates. Could the authors include additional Methods details and plots, such as:

- How many biological or technical replicates were conducted for each screen?

What is the correlation between biological and technical replicates, at the level of individual pgRNAs and at the level of aggregated element scores (e.g. from RELICS)?

- For genes with both mCherry and GFP reporter in different alleles, do the results correlate between the sorting of each reporter?

What approximately is the distribution or range of efficiencies of different pgRNAs in this system?

- For individual elements of interest, could you show e.g. in locus plot the scores observed for each pgRNA spanning the element?

- Line 91: What does probability score 0.1 mean? Could you summarize why this is a good threshold and what validations have been done to select the threshold? How will the number of identified enhancers change based on this threshold? Does it make sense to have the same threshold for different screens and libraries done on 4 genes?

2. The paper claims to find “hidden enhancers that do not have typical epigenetic features”. This claim should ideally be supported by additional experimental evidence, given other possible explanations such as:

- Unexpected effects from genetic deletions
- Deletions of RNA or splicing elements in the gene body (e.g., E3 in Fig 2c is in a 3' UTR)
- Deletion of repetitive sequences that may be marked by histone modifications but where reads cannot be uniquely aligned
- CTCF binding sites (which would be accessible but not necessarily marked by other histone modifications)
- Indirect effects of an element regulating one gene which then affects the expression of the reporter gene in trans

At the very least, the authors should:

- Provide genomic locus plots showing ATAC-seq, other chromatin marks such as H3K27ac, and Hi-C 3D contacts (these are mentioned but not shown) at each of the 4 loci and at the identified ‘enhancers’ (e.g., in Figs 2a,c, 3a, and/or in supp), and ideally a genome browser link for interactive study

- Adjust the language (e.g., “regulatory elements” might be more appropriate for the set of regions identified in the screens, versus “enhancers”)

- Tone down the claims in the Discussion (e.g. by discussing these other possible explanations)

- Please mention in the Main Text whether reduced expression of any of the 4 genes is expected to impact neuronal differentiation and the dynamics of their expression over the course of differentiation. I am wondering how these aspects affect interpretation of the sorting strategy (i.e., are you expecting that some deletions will reduce neural differentiation and therefore lead to indirect effects on genes whose expression is supposed to be up-regulated during differentiation)

3. Genetic deletions with Cas9 or 2-guide Cas9 can lead to unexpected outcomes, such as inversions of the deleted sequence, loss of chromosome arms, scars at the cut site that leave a band that appears to be the length of the unedited allele on a PCR gel, etc. Specific questions:

- Have the authors quantified the frequency of such effects in their hiPSC system, and estimated how it might affect the screen results?

- What genotyping strategies were used for ensuring such effects were not present in clonal cell lines? E.g., is the ‘wild-type’ allele in the heterozygous deletion clones indeed unedited?

4. Could the authors redefine the phenomenon of “ACE” in the Discussion, and explain more extensively to previously observed phenomena? It seems to me that there are many other similar examples that have been described (e.g., many loci previously found to have negative feedback loops would be expected to behave this way, and heterozygous knockout mice can end up with more than 50% of protein expression). A few specific examples that come to mind include CHD2 (also a chromatin regulator that binds near its own promoter to down-regulate itself) and the many RNA splicing factors known to auto-regulate themselves e.g. through alternative splicing of poison cryptic exons. It would be helpful to discuss and

compare to other such examples.

5. Resolution of the analysis

It appears that RELICS sometimes calls very large regions (e.g., 65,981 bp in the gene body of MECP2) as a single regulatory element. Why is the resolution at this gene different than others where 'gene-body enhancers' are called? Again, viewing the scores for all individual pgRNAs would be helpful here to understand what is going on. The dense tiling (15-20x coverage for each nucleotide) is impressive. Did it pay off? Did it help discover regulatory elements that otherwise would be missed? Commenting on this would strengthen the technical contribution of this paper and motivation for this particular design.

6. Line 203: "the partial reduction of SIN3A expression from the promoter deletion allele may not be sufficient to induce ACE..." The data in Fig 4c does not seem to support the hypothesis, esp. since 4b shows compensation happening as early as Day 2, when the knockdown effect for enhancer and promoter seems to be identical (Fig. 4c). Could you explain further?

Minor points:

- Fig 1b: The layout of the legend was confusing — I thought at first that the colored words corresponded to x-axes or labels for each of the 4 panels, rather than being a legend that applies to all 4 panels. Changing the layout would improve clarity.
- Fig 1b: Filtering to $\log_2FC > 0$ seems inappropriate here. Would it not be better to show the p-value distribution for all pgRNAs?
- Fig 1b: Please define "Test pgRNAs" (blue) in legend
- Fig 2a. Please label the site of the reporter insertion knock-in ... how close is it to E3?
- "While copy number loss variants overlapped with SIN3a Enhancers identified from clinical samples are likely benign" — Could you clarify whether there is clinical evidence that the enhancers are benign, or is this an inference from the regulatory model proposed?
- Generation of deletion clones: Could you clarify how the FACS was done (were clones with low reporter expression single-cell sorted, then genotypes, or was the single-cell sorting + genotyping + clone selection done without regard to reporter expression)? How much time elapsed between single-cell sorting and reporter readout? How many clones were studied in each case (I see two clones in most figures), and how were these two clones selected?
- Fig. 3d,e — Which enhancer deletion does this represent? Please include in Figure legend, not just the main text
- Line 60: "tilling" → "tiling"
- Line 83: Is Cas9 expressed the whole time or Cas9 is background silenced after differentiation? So the effective editing time is 1 week or 3 weeks?
- Line 113: Can you tell this enhancer is a neuronal-specific enhancer from chromatin features such as K27ac?
- Line 126: What is the orientation of this CTCF motif? Taking into account the relative position of this enhancer to the target gene, does it make sense that disrupting the CTCF affects gene expression?
- The paragraph on Line 132 about analysis of ClinVar CNVs seems out of place. Perhaps consider moving this to the end of manuscript, or commenting more about the results for the 4 genes under study?
- Line 222: Could you remind reader in this paragraph that SIN3A is a transcriptional repressor
- Line 223: The locus plot showing the Sin3A ChIP-seq is helpful. Could you please also comment on how many expected binding sites there are?
- Fig. 2G, 3B: From Fig.1, you discovered more than a dozen regulatory elements for SIN3A. Could you please clarify why it is plausible that deleting any one of these distal regulatory elements E2-E5, which are more than 600 kb away from the promoter, had such a strong effect on gene expression close to a complete knockdown?
- Line 237: "Our ACE model can also explain the haploinsufficiency of SIN3A236 for the Witteveen-Kolk syndrome (WITKOS) patients with large deletions of the entire SIN3A locus 237 including the SIN3A promoter...while copy number loss variants overlapped 238 with SIN3A enhancers identified from clinical samples are likely benign." I'm confused by the logic here – why would losing the whole locus lead to different phenotype as losing the distal enhancer? For the current experiment, the fact that deleting the promoter can't sufficiently induce ACE can be explained by alternative TSS, but losing the whole locus should surely induce ACE according to your model?
- Fig 2J: For this one and the three additional ones in the supplementary figure: the total editing rate is about 50%, 80%, 45%, and 50%. And these are from the cells that have an effect. Then what about the rest of the sequences that don't have

intended edits? Why do they cause a decrease in the expression of the gene?

- Fig. 4A: Use the diagonal line used in Fig2G to help with visualization
- Extended Data Figure 6: Why is the enhancer deletion effect biased toward GFP? Not seen for the promoter deletion

Reviewer #3

(Remarks to the Author)

In this manuscript, Ren et al. utilized a high-resolution genetic screening approach to identify functional enhancers of four neuropsychiatric risk genes—APP, FMR1, MECP2, and SIN3A—in iPSC-induced excitatory neurons. Among the several novel enhancers validated through this CRISPR tiling study, the authors focused on the allelic compensation effects (ACE) of targeting SIN3A enhancers. A sophisticated allelic reporter system, tagging SIN3A with EGFP and mCherry on separate alleles, was employed to characterize the transcriptional regulation of each allele upon enhancer deletion.

The results indicate that the loss of SIN3A expression from one allele due to enhancer deletion resulted in increased transcriptional activity on the sister allele, thereby maintaining overall SIN3A expression. The authors propose that this enhancer-mediated ACE is essential for the maintenance of SIN3A—a haploinsufficient gene—and that similar mechanisms might regulate other dosage-sensitive genes.

While the experiments in this study are well-designed and the CRISPR tiling screens provide an unbiased evaluation of the selected gene loci, several issues require further characterization:

1. Fig. 1b: It would be more informative to present the CRISPR screen results using volcano plots, where the x-axis represents log₂ fold-change and the y-axis represents $-\log_{10}(P \text{ value})$.
2. MECP2-E6: MECP2-E6 is characterized as a neuron-specific enhancer. However, the absence of an EGFP-depleted cell population in MECP2-E6-targeted iPSCs (Fig. S5) could also be caused by a lower CRISPR deletion efficiency at the MECP2-E6 locus in iPSCs. The authors should address the limitations of dual-sgRNA deletion efficiency and other alternative possibilities in discussions.
3. APP-E3: Targeting APP-E3 (Fig. S6) had a stronger effect on the EGFP allele (7.41% and 5.72%) than on the mCherry allele (0.79% and 0.7%). This unique phenomenon was not observed at other loci. Could the authors explore whether allelic-specific sequences at the APP-E3 locus affect CRISPR targeting efficiency on the mCherry allele?
4. Line 197–199: “The observed dynamic rate of ACE after enhancer deletion suggests that ACE is more potent as SIN3A expression approaches the level that triggers haploinsufficiency after day 5.” This observation implies that, in addition to active ACE, survival selection might contribute to the enrichment of higher SIN3A-expressing cells over time. This possibility should be considered.
5. Line 201–204: “Long-read RNA-seq data revealed SIN3A transcription from two TSSs, and we only deleted the promoter of the major SIN3A transcript (Extended Data Fig. 11c,d). Thus, the partial reduction of SIN3A expression from the promoter deletion allele may not be sufficient to induce ACE.” This hypothesis should be tested by deleting both SIN3A TSSs to determine whether ACE is observed under complete loss of SIN3A expression from one allele.
6. Line 218–219: “These results suggest ACE, once established, cannot be reversed by increasing SIN3A expression.” The ectopic expression of SIN3A-Pr > SIN3A-P2A-BFP failed to reverse SIN3A enhancer ACE (Fig. 4f), raising concerns about whether the “dosage effect” is the primary contributor to ACE. To address this, overexpressing SIN3A using a viral promoter (e.g., CMV) might bypass potential autoregulatory feedback affecting SIN3A-Pr > SIN3A-P2A-BFP expression. Additionally, the authors should compare the effects of ectopic expression of SIN3A-P2A-BFP before versus after SIN3A enhancer deletion to clarify the proposed hypothesis.
7. ChIP-qPCR: To support the proposed model summarized in Fig. 5e, ChIP-qPCR analysis of SIN3A binding at SIN3A promoter loci should be performed.

Reviewer #4

(Remarks to the Author)

Version 1:

Reviewer comments:

Reviewer #1

(Remarks to the Author)

I would like to thank the authors for addressing my comments carefully.

The only response I find a bit worrying is in response to my initial comment on the previous lines 91-104 where I asked for a comparison to previously established regulatory elements. I proposed there to use a human specific atlas of such regulatory elements generated by a large meta-analysis of human brain epigenome data. The authors reply that they did not use this resource for their comparisons, as it would contain data sets that would not be derived from human samples. This is incorrect (only some human regulatory elements in that study have been tested in animal models, but are still human sequences). Hence this incorrect reply makes me suspiciously wonder whether the authors might have tried to use that alternatively proposed atlas and might have found results that did not fit into their expectations and might have therefore chosen alternative sources where the results might have fit their expectations better? The authors might want to comment on that, to prevent the impression of any potential selection bias introduced here.

Reviewer #2

(Remarks to the Author)

The reviewers have revised the manuscript substantially and made numerous improvements.

A few additional edits would be sufficient to address my previous major points:

- For the Extended Data Figure 4 (reproducibility of CREST-seq screens), please show full scatterplot of RELICS probability scores between replicates 1 and 2 for each screen (e.g., each point = one segment, 'hit' segments colored) rather than simply reporting the Pearson correlation, because good Pearson correlations can be achieved by e.g. single outliers. Also please report the Pearson correlation of RELICS probability scores considering just the 'hits'. I also think it is necessary to show scatterplots at the level of individual pgRNAs, not just a PCA plot.
- Thanks for adding the genome browser plots, very helpful. One small addition that would make these even better would be to show in the browser track which regions are covered by pgRNAs (are there any gaps in coverage due to repetitive sequences or other design choices/limitations?)
- In the Response, the authors wrote: "In our screens, we identified functional elements based on their impact on reduced target gene expression when perturbed. Because these elements positively regulate transcription, we believe the term "enhancer" is appropriate for describing the functional elements we identified." I disagree with this definition, but, at the very least, the authors should include their own definition early in the manuscript.
- This sentence of the response should be included in the Methods or Supp Figure legend, because it is a critical detail for interpreting the expression effects observed in the deletion clones: "To generate deletion clones, we sorted the single cells with reduced expression of the reporter into 96-well plates with one cell per well. It takes about 3-weeks between single-cell sorting and genotyping. We randomly choose two clones for each analysis."

Reviewer #3

(Remarks to the Author)

The authors have addressed the previous concerns by including additional data and analyses, which have considerably strengthened their claims. The manuscript is now substantially improved, and I have no further comments.

Reviewer #4

(Remarks to the Author)

Version 2:

Reviewer comments:

Reviewer #1

(Remarks to the Author)

the authors have sufficiently addressed all my comments, and I would like to congratulate them on this very nice study.

Reviewer #2

(Remarks to the Author)

The authors have addressed all of my remaining comments. Please publish!

We thank all reviewers for their constructive comments, which helped improve our manuscript significantly. Motivated by their suggestions, we have added new analyses and additional experiments as requested. In addition, we removed the analysis of ClinVar CNVs, as two reviewers noted that it does not align well with the objectives of this study. Please refer to our point-by-point response to each of the reviewers' comments. Changes to the manuscript are highlighted in blue.

Reviewer #1 (Remarks to the Author)

Ren et al use a CRISPR based tiling screen to identify functional enhancers of neuropsychiatric risk genes, and obtain evidence of transcriptional allelic compensation effects. This is a very interesting study that I read with great pleasure, and which will deserve publication in a good journal. However prior to that, there are a number of (significant) issues to be addressed:

Comments (in the order of appearance in the manuscript):

-the title indirectly implies to me that functional enhancers of many neuropsychiatric risk genes are assessed; at the end, those are “only” four genes. I think the authors should therefore tone down a bit the title that might give a wrong impression for the reader what is to be expected from the paper. Same applies to the ACE that only is found at the *SIN3A* locus, and not a general phenomena of the other loci investigated. APP is involved in Alzheimer's disease. Although Alzheimer's can have neuropsychiatric symptoms, I am not completely sure that in general Alzheimer is considered a neuropsychiatric disorder? Rather a neurodegenerative one. So the authors should also consider their language use on this point.

We thank Reviewer #1 for the comments. We agree that only four genes were tested in this study, and changed the title to “CRISPR tiling deletion screens reveal functional enhancers and allelic compensation effects (ACE) on *SIN3A* transcription”. We also change our description in the main text from “Genetic analyses have identified numerous neuropsychiatric risk genes” to “Genetic analyses have identified numerous risk genes for neuropsychiatric and neurodegenerative diseases” (line 42).

-OMIM numbers should be used where in the manuscript known disorders / syndromes are first mentioned (for example in the second paragraph of the introduction).

We thank Reviewer #1 for bringing up this point. We added the OMIM numbers of the mentioned disorders/syndromes in the revised manuscript (lines 46-56) except for Fragile X-associated neuropsychiatric disorders (FXAND), which does not have a designated OMIM number.

-around line 74: it should probably be specified that the WTC11 cell used is a male cell line, hence only requiring to tag a single copy of the X-linked genes that are studied.

We thank Reviewer #1 for bringing up this point for clarification. We updated this information in the revised manuscript (line 75).

-around line 80: did the authors check whether there was any specific gRNA depletion at the iPSC cell stage, that was no longer present in the induced neurons, which could point to any deletion effects impacting on the differentiation of the induced neurons and might therefore be selected against?

We thank Reviewer #1 for bringing up this clarification point. To assess whether any pgRNAs were selectively depleted at the iPSC stage due to effects on neuronal differentiation, we examined the representation of pgRNAs in the control libraries of 2-week neuron libraries. We found that over 99% of the designed pgRNAs were retained. This high recovery rate, along with our high coverage design (15× or 20×) confirms that our screening strategy is robust and not confounded by early depletion of pgRNAs that might impair differentiation.

Target gene	The number of recovered pgRNAs	The number of designed pgRNAs	Recover rate
APP	10,747	10,788	99.62%
FMR1	16,684	16,705	99.87%
MECP2	17,233	17,293	99.65%
SIN3A	14,361	14,471	99.24%

-line 91-104: I appreciate the detailed analysis of the identified enhancers vs various epigenome data sets. It might however further strengthen the data to also compare the identified enhancer to previously established resources for brain related enhancers, to get a glance how many were previously unidentified. For example, the data could be compared to those in the atlas generated in PMID: 34663447 or other related work. Especially for the ones not associated with enhancer associated chromatin marks, it might be interesting to see whether those sequences are actually marked with such marks at other brain developmental stages but just not in the cell type that is being investigated (some kind of priming?)

We appreciate the reviewer's suggestion and conducted additional analysis. We did not use the atlas generated in PMID: 34663447, because it includes data from multiple different resources and some of them are not derived from human samples. Instead, we used three datasets generated from human brain samples covering an age range of 89 days (estimated postconceptual age) to 89 years¹⁻³. By comparing our identified

enhancers with human brain candidate *cis*-regulatory elements (cCREs) in these three studies, we found that canonical enhancers showed a significantly higher overlap with human brain cCREs compared with hidden enhancers (82.1% vs 18.2%). The result suggests that hidden enhancers tend to remain hidden across multiple cell types. We added these results into the manuscript (lines 105-108, Figure 1e).

	Overlap with cCREs identified from human tissues ¹⁻³
Canonical enhancers	23/28=82.1%
Hidden enhancers	2/11=18.2%

Figure 1e. Overlap between identified enhancers and cCREs identified from human brain samples. *P* value was determined using the two-tailed Fisher's exact test.

-line 132-148: I can see the motivation (and appreciate) to show some “clinical utility” of the findings from the screen, by employing CNV enrichment analysis. In its present state however, the analysis is meaningless. Most importantly, if the hypothesis is true (and I don't have doubts about that) that genetic variants of the functional identified enhancers can contribute to disease, this would certainly be only true for deletions that remove an enhancer, or for other point mutations / small alterations that disrupt the enhancer itself (for example interfering with a TF binding site). Hence, in their CNV analysis, the authors should only focus on deletions (e.g., for CNVs that are duplications, it is by no means clear from usual diagnostic SNP-array data whether such duplications occur in *cis* / in tandem, or might mess up the 3D organization or even be integrated in completely different loci) as for all other CNVs it cannot be just assumed that there is a *cis*-regulatory effect. Also the enrichment analysis from line 140 onwards seems to be completely disconnected from the 39 functional enhancers that the authors identify, now moving to cCREs (identified from where?) identified to 355 disorders? There is already quite some literature that shows enrichment of deletions over potential regulatory elements possibly contributing to disease, so adding this unrelated analysis and concluding with line 147/148 is a bit too much selling of the importance of the findings in my opinion. It would be better if the authors would search for specific deletions overlapping specifically the functional enhancers identified, that could be found in yet undiagnosed patients. There are already great resources available to this (for example the Genomics England 100,000 genome data, data from the GREGOR consortium, and plenty of clinical geneticists likely be excited to collaborate on this as

this might help to resolve current “missing heritability”). Also, alternatively, the authors could use publicly available genome data from gnomADv4 to show that their functional enhancers are depleted for variants in the healthy population, further emphasizing their potential role in disease.

We appreciate the reviewer’s constructive advice. Following the reviewer’s suggestion, we investigated whether the enhancers identified in our screen overlap with deletions reported in external variant datasets. We found none of the 39 enhancers we identified overlap with any GREGOR variants, and 15 of the enhancers overlapped with deletions from gnomADv4. Since Reviewer #2 pointed out that this analysis was somewhat out of scope for the current study, we have decided to remove this section from the revised manuscript for clarity and focus.

-line 150-219: the observation of ACE at the SIN3A locus is highly interesting, and the authors use elegant experiments trying to understand this mechanism. As the authors also mention, my feeling would be that this effect is likely caused at this locus given the essentiality of SIN3A and its autoregulation, which might not tolerate reduced expression. The authors already elegantly check that once the ACE is established in their enhancer deletion clones, overexpression of exogenous SIN3A cannot restore this. I think one experiment that is missing and which should be added would be the reverse. E.g., can the SIN3A exogenous overexpression first be established in the double reporter background, and subsequently the enhancer be deleted. In that case, you would expect that ACE would not occur, given that the SIN3A dosage problem for the cell is solved by the exogenous expression. Adding this experiment could more securely nail down that the observed ACE is due to the essentiality of the gene. In general, if the authors would like to make more broad statements about ACE (which I feel is the case given the manuscript title), my advice would be to chose another essential gene and perform the same analysis (but the simpler solution would be to tone down on the claims and make sure that it is clear that this so far is a locus specific SIN3A observation). Alternatively, the authors should look in literature for other examples and discuss them. To my mind come studies from Pablo Navarro and Ian Chambers looking at the auto-regulation of the NANOG locus in mouse ES cells (PMID: 23178592 and perhaps also later work of them). I am not completely sure as it has been a while that I was working in that field, but I think with reporters at that locus very similar observations had been made when altering one of the upstream NANOG enhancers.

We thank the reviewer for the insightful comments and thoughtful suggestions. Following reviewer’s advice, we generated SIN3A-EGFP/mCherry iPSCs with a *SIN3A* overexpression background by introducing a CAG promoter controlled SIN3A-P2A-BFP transgene via lentivirus infection. The overexpression of *SIN3A* was confirmed by RT-qPCR (Extended Data Fig. 13d).

Extended Data Figure 13d. RT-qPCR showing *SIN3A* expression.

Next, we deleted the *SIN3A* enhancer (*SIN3A-E4*) in the *SIN3A* overexpressed iPSCs and differentiated enhancer deletion clones, along with matched controls (#1: Cells with CAG-*SIN3A* overexpression; #2: *SIN3A*-EGFP/mCherry reporter cells; #3: *SIN3A*-E4 deletion clones isolated from #2 *SIN3A* reporter cells). We then performed flow cytometry analysis in iPSCs and neurons to quantify endogenous *SIN3A* allelic expression. By calculating the compensation index (CI), defined as the ratio of EGFP or mCherry signal (from the compensation allele) in enhancer deletion clones relative to control cells. We observed that *SIN3A* overexpression significantly reduced the CI compared to cells without *SIN3A* overexpression, indicating that exogenous *SIN3A* overexpression suppresses the ACE response. We have incorporated these findings into the revised manuscript (lines 207-214, Figure 4f). Furthermore, we found that ACE can also happen to *NANOG* from a recent study of *NANOG* enhancer in hESCs⁴. Specifically, the authors kindly provided us their original data for the allelic ddPCR data on *NANOG* expression in *NANOG* e1 deletion heterozygous clones. Deletion of *NANOG* e1 enhancer led to downregulation of *NANOG* from the same allele, and increased expression of *NANOG* from the other allele. This example confirms that ACE is not restricted to *SIN3A*. Notably, *NANOG* is one of the candidate genes in our predicated candidate gene list. We add this new result to lines 265-268 and Extended Data Figure 16.

Figure 4f. *SIN3A* overexpression represses ACE. The ACE in *SIN3A*-E4 deletion clones isolated from *SIN3A*-EGFP/mCherry reporter cells with and without CAG promoter controlled ectopic *SIN3A* expression. C1 and C2 indicate two independent clones for each genotype. Control cells are *SIN3A*-EGFP/mCherry reporter cells, with (CAG>*SIN3A*) or without *SIN3A* overexpression (WT). The compensation index (CI) is defined as the ratio of EGFP or mCherry signal (from the compensation allele) in enhancer deletion clones relative to control cells. *P* values were determined using the two-tailed two-sample t-test.

Extended Data Figure 16. Allelic expression of *NANOG* in *NANOG* e1 enhancer deletion clones. a,b, ddPCR results of *NANOG* allelic expression in *NANOG* e1 enhancer deletion clones. Data were reanalyzed from Yan *et al*⁴.

-line 230: the drawn conclusion is wrong: “Both P1 and P1+P2 promoter reporters exhibited a significant increase of promoter activity when endogenous *SIN3A* expression is reduced by *SIN3A* shRNA (Fig. 5b-d). Thus, the *SIN3A* promoter can counteract allelic enhancer deletion-induced downregulation by increasing its transcriptional activity.”

All what the authors show is that the autoregulation is (as expected) mediated by the SIN3A promotor. The authors simply test promotor activity upon shRNA knockdown of SIN3A in these experiments. That is independent from their earlier findings on the enhancer deletion. The authors should thus rephrase their conclusions accordingly. If the authors want experiments that support their current conclusion, they should look at promotor output in the cells having the SIN3A enhancer deletion and not do the indirect shRNA knockdown/transfection in wild type cells.

We thank the reviewer for this thoughtful comment. To test whether *SIN3A* promoter output changes can be affected by SIN3A levels, we tested the *SIN3A* promoter in shRNA-mediated *SIN3A* knockdown cells and wild-type cells. As the reviewer pointed out, our shRNA knockdown experiments specifically test SIN3A promoter activity under reduced SIN3A levels, independent from enhancer deletion. In heterozygous cells with the *SIN3A* enhancer deletion, we observed elevated promoter output on the non-deleted allele and reduced output from the deleted allele (Figures 2g, 3e, 3f). Our purpose is to show that promoter activity can be regulated by total SIN3A level. However, testing endogenous promoter activity in these enhancer-deleted cells would not be informative, since allelic compensation (ACE) results in total SIN3A expression levels comparable to wild-type cells (Fig. 3f). To further support our conclusion, we performed CHIP-qPCR experiments and showed the binding of SIN3A at *SIN3A* promoter region (Fig. 5e).

Following reviewer's advice, we deleted the sentence "Thus, the *SIN3A* promoter can counteract allelic enhancer deletion-induced downregulation by increasing its transcriptional activity." in our manuscript.

Figure 5e. SIN3A binding at the *SIN3A* promoter region assessed by CHIP-qPCR. Data are shown as mean \pm SD. *P* values were determined using a two-tailed two-sample t-test. Dots represent individual biological replicates. R1: Region 1; R2: Region 2; R3: Region 3.

-line 244-262: again, this analysis is a bit distracting. I can see why the authors perform this, but once again in the current state the analysis doesn't say much more than that there are plenty of genes encoding for regulatory factors that bind to their own promotor, that are widely expressed, dosage sensitive and therefore like also enriched for being disease relevant genes. However, the analysis does not say for any of these genes, that ACE is happening (e.g., that if the authors would perform the same enhancer deletion for the other loci, the same upregulation of the other allele would happen). Hence I really don't like this effort of overselling the conclusions, and I think the authors should significantly tone down on them. This is not to criticize their experimental findings, as I think the data is very interesting and solid but they should just be described as they are and deliberated from the sauce of overselling in my opinion.

We thank the reviewer for the constructive comment. Following reviewer's advice, we reanalyzed the raw data from the published study of allelic enhancer deletion of *NANOG* in hESCs⁴. Here we show that downregulation of *NANOG* from the e1 deletion allele are coupled with the upregulation from the intact allele (Extended Data Fig. 16), confirming that the ACE is not limited to *SIN3A*.

Extended Data Figure 16. Allelic expression of *NANOG* in *NANOG* e1 enhancer deletion clones. a,b, ddPCR results of *NANOG* allelic expression in *NANOG* e1 enhancer deletion clones. Data were reanalyzed from Yan *et al*⁴.

In addition, we further toned down our conclusion to “These candidate genes suggest that ACE could be a widespread gene regulatory mechanism for dosage-sensitive genes.” (lines 265-266).

In summary, I really do like the majority of the wet lab experiments performed and are therefore supportive of publications once my concerns are addressed upon revision. Some of the in silico analysis is distracting and not supportive for the case that the authors want to make. The manuscript would benefit from toning down of some of the overselling. The revision in my opinion should focus on 1) the added experiment with first exogenous *SIN3A* overexpression followed by enhancer deletion to disentangle

causality of the observed ACE ; 2) alternative choices to further strengthen the generalizability of the ACE to other loci ; 3) toning down on the slightly overselling of findings to make sure that the conclusions are really supported by the presented data.

We thank the reviewer for their valuable comments and suggestions. We have conducted additional experiments and analyses, and revised the manuscript accordingly. We hope these changes satisfactorily address the reviewer's concerns.

Hope this helps, best wishes, Stefan Barakat

Reviewer #2 (Remarks to the Author):

This study identifies noncoding DNA elements that regulate 4 dosage-sensitive and disease-relevant genes in iPSC-derived excitatory neurons. Using 2-guide CRISPR/Cas9 tiled deletion screens, the authors identified 39 cCREs that affect the expression of one of these genes. They dissect the functions of several of these functional elements and discover that deletion of an element that regulates SIN3A leads to a strong decrease in expression from the SIN3A allele in cis and a compensatory in expression from the allele in trans. The authors show that SIN3A, a transcription repressor, directly binds its own promoter, providing a potential molecular mechanism for this phenomenon.

Identification of regulatory elements for dosage-sensitive neuropsychiatric genes is an important and interesting goal. Discovery of very long-range regulatory elements for SIN3A is supported by nice analysis of heterozygous deletion clones with allele-specific expression readouts. The location of these regulatory elements is surprising (overlapping other genes, located >600Kb away and with a dozen intervening genes). The CRISPR screening tool applied here (CREST-seq) has been applied previously by the authors, but this represents the most extensive application of the tool to my knowledge to date, and has the potential to yield important technical lessons on what this type of genetic deletion screening can find.

Overall, I would support publication if the following major points can be addressed:

Major points:

1. Technical quality and reproducibility of CREST-seq experiments. I am having trouble assessing the overall robustness of the screen and downstream analysis. The validation

experiments for many individual elements are very helpful but more information is needed, for example to assess false positive or false negative rates. Could the authors include additional Methods details and plots, such as:

- How many biological or technical replicates were conducted for each screen?

We did two biological replicates for each screen as shown in lines 85-86.

What is the correlation between biological and technical replicates, at the level of individual pgRNAs and at the level of aggregated element scores (e.g. from RELICS)?

According to reviewer's suggestion, we performed Principal Component Analysis (PCA) at individual pgRNAs level to assess the correlation between biological replicates. In each screen, the primary source of variance was the separation between control cells and sorted cells with reduced expression of EGFP/mCherry reporters (Extended Data Fig. 4a). The clustering patterns confirmed the reproducibility of the screen and the robustness of the reporter-based selection strategy.

At the element level, we analyzed each biological replicate with RELICS, and checked the Pearson correlation of the RELICS probability score between replicates. We found a high correlation of RELICS score in each screen (Extended Data Fig. 4b). These results are now included in Extended Data Figure 4a,b.

Extended Data Figure 4. The reproducibility of CREST-seq screens. **a**, PCA analysis of CREST-seq screens. **b**, Pearson correlation of the functional sequence probability scores from RELICS analysis between biological replicates for *APP*, *FMR1*, *MECP2* and *SIN3A* screens, and between EGFP and mCherry alleles for *APP* and *SIN3A* screens.

- For genes with both mCherry and GFP reporter in different alleles, do the results correlate between the sorting of each reporter?

To check the correlation between two reporters, we analyzed the *APP* and *SIN3A* screens using the data from EGFP allele and mCherry allele individually. RELICS splits the target region into segments and calculates the RELICS score for each segment. We plotted the RELICS score from EGFP allele and mCherry allele, and found a high correlation between two reporters for both *APP* ($r=0.99$) and *SIN3A* ($r=0.95$) screens. This result is now included in Extended Data Figure 4b.

What approximately is the distribution or range of efficiencies of different pgRNAs in this system?

We thank the reviewer for bringing up this question. To check the efficiency of pgRNAs in each library, we used Rule Set 2 score⁵, a commonly used method for estimating sgRNA efficiency. We assessed efficiency separately for left sgRNA, right sgRNA, and sgRNA pair (the average of left and right sgRNAs). These libraries showed similar sgRNA efficiency distributions, with median Rule Set 2 scores around 0.6 (Extended Data Fig. 2d), comparable to values reported for effective sgRNAs in prior CRISPR screens (Fig. 4e in PMID: 26780180). In addition, we designed multiple sgRNA pairs (15× or 20× coverage) for each target region to compensate for potential low editing efficiency with some sgRNAs.

Extended Data Figure 2d. The sgRNA efficiency (Rule Set 2 score) of sgRNAs in each pgRNA library.

	APP		
	Left sgRNAs of each pair	Right sgRNAs of each pair	sgRNA pairs
Median	0.57139	0.57171	0.5623
Mean	0.55378	0.55396	0.5539

	FMR1		
	Left sgRNAs of each pair	Right sgRNAs of each pair	sgRNA pairs
Median	0.57558	0.57552	0.5665
Mean	0.55733	0.5579	0.5576

MECP2			
	Left sgRNAs of each pair	Right sgRNAs of each pair	sgRNA pairs
Median	0.57745	0.57787	0.5706
Mean	0.5635	0.56446	0.564

SIN3A			
	Left sgRNAs of each pair	Right sgRNAs of each pair	sgRNA pairs
Median	0.58158	0.5791	0.5727
Mean	0.56662	0.5653	0.566

- For individual elements of interest, could you show e.g. in locus plot the scores observed for each pgRNA spanning the element?

We thank the reviewer for the suggestion. The functional elements in each screen were identified by using RELICS. RELICS splits the region of interest into segments. It then iteratively places one functional sequence at a time, while considering all previously placed functional sequences. RELICS provides scores for each segment instead of for each pgRNA. We added a track to show the RELICS scores in the locus plot and a genome browser session

(https://epigenomegateway.wustl.edu/browser2022/?genome=hg38&sessionFile=https://shen-xren.s3.us-west-1.amazonaws.com/CREST-seq_NC/eg-session-MURg2uS0Q-ce92ef10-72ad-11f0-b5d2-a94bbd38dced.json).

This information is added to Data availability statement. In addition, we added genome browser snapshots to show the RELICS scores in each locus (Extended Data Fig. 6).

- Line 91: What does probability score 0.1 mean? Could you summarize why this is a good threshold and what validations have been done to select the threshold? How will the number of identified enhancers change based on this threshold? Does it make sense to have the same threshold for different screens and libraries done on 4 genes?

We thank the reviewer for raising these important questions. RELICS divides the target region into segments and calculates the similarity between each segment and positive controls based on the signal in CRISPR screens. The probability score represents the likelihood that an element functions as a regulatory element compared with the positive control elements. A score of 0.1 is the default cutoff for identifying functional elements in RELICS.

After analyzing the screens using this default cutoff, we merged the adjacent functional sequences and calculated the median RELICS score for each merged DNA fragment using bedtools (v2.26.0). Then, we checked the overlap between the merged DNA fragment and chromatin features (the features used in Figure 1d). As shown in Figure R1, fragments with scores between 0.1-0.2 had the highest percentage (54%) lacking chromatin features we examined, compared to other groups. This analysis suggests that the default cutoff of 0.1 is not stringent and may include false positives. To reduce this risk, we applied a more stringent cutoff, retaining only merged fragments with a median RELICS probability score >0.2 and at least two functional sequences as enhancers.

We further performed extensive validation for enhancers spanning a broad range of probability scores (0.3 - 0.9).

As described in the method section “Analysis of CREST-seq screens”, we used pgRNAs overlapping 5’TUR and exons of each target gene as positive controls for each screen. Because functional enhancers in each screen were identified relative to these internal positive controls, applying the same probability score threshold for different screens is appropriate and ensures comparability.

Figure R1. The RELICS probability score of merged DNA fragments.

2. The paper claims to find “hidden enhancers that do not have typical epigenetic features”. This claim should ideally be supported by additional experimental evidence, given other possible explanations such as:

- Unexpected effects from genetic deletions
- Deletions of RNA or splicing elements in the gene body (e.g., E3 in Fig 2c is in a 3’ UTR)
- Deletion of repetitive sequences that may be marked by histone modifications but where reads cannot be uniquely aligned

- CTCF binding sites (which would be accessible but not necessarily marked by other histone modifications)
- Indirect effects of an element regulating one gene which then affects the expression of the reporter gene in trans

At the very least, the authors should:

- Provide genomic locus plots showing ATAC-seq, other chromatin marks such as H3K27ac, and Hi-C 3D contacts (these are mentioned but not shown) at each of the 4 loci and at the identified 'enhancers' (e.g., in Figs 2a,c, 3a, and/or in supp), and ideally a genome browser link for interactive study

We appreciate the reviewer for comments and suggestions. We added a genome browser session for displaying the epigenetic features for the 4 loci, including gene expression (RNA-seq), ATAC-seq, chromatin marks, cCREs annotation from human brain samples¹, H3K4me3 mediated PLAC-seq, RELICS scores, identified functional enhancers from each screen, and sgRNAs used for validation experiments. This genome browser session enables readers to explore this study in detail (https://epigenomegateway.wustl.edu/browser2022/?genome=hg38&sessionFile=https://shen-xren.s3.us-west-1.amazonaws.com/CREST-seq_NC/eg-session-MURg2uS0Q-ce92ef10-72ad-11f0-b5d2-a94bbd38dced.json).

In addition, we added genome browser snapshots of the loci containing the enhancers we validated individually (Extended Data Fig. 6).

- Adjust the language (e.g., "regulatory elements" might be more appropriate for the set of regions identified in the screens, versus "enhancers")

We thank the reviewer for the suggestion. In our screens, we identified functional elements based on their impact on reduced target gene expression when perturbed. Because these elements positively regulate transcription, we believe the term "enhancer" is appropriate for describing the functional elements we identified.

- Tone down the claims in the Discussion (e.g. by discussing this other possible explanations)

We thank the reviewer for suggesting possible explanations for hidden enhancers. We carefully checked the eleven hidden enhancers we identified. Three of them are located in the intergenic regions. Seven of them are located in the introns, but far away from the upstream and downstream splicing sites. They are less likely to affect splicing.

Enhancer_label	Hidden_enhancer_annotation
APP_E1	Intergenic region
APP_E4	Located in APP intron, 3.5 kb from the upstream and 38 kb from the downstream splicing site
MECP2_E5	Located in MECP2 intron, 2.5 kb from the upstream and 54 kb from the downstream splicing site
MECP2_E8	Located in MECP2 intron, 17 kb from the upstream and 41 kb from the downstream splicing site
MECP2_E11	Located in MECP2 intron, 45 kb from the upstream and 10 kb from the downstream splicing site
MECP2_E12	Intergenic region
SIN3A_E1	Located in EDC3 intron, 2.5 kb from the upstream and 1.5 kb from the downstream splicing site
SIN3A_E3	CYP1A2 exons, CYP1A2 is not expressed in neurons with RPKM = 0
SIN3A_E14	Located in PPCDC intron, 11 kb from the upstream and 1.5 kb from the downstream splicing site
SIN3A_E17	Intergenic region
SIN3A_E18	Located in TMEM266 intron, 41.4kb from the upstream and 32.5 kb from the downstream splicing site

Among these hidden enhancers, we validated two (MECP2_E8, SIN3A_E3) with pgRNAs-mediated deletion, including MECP2_E8 and SIN3A-E3. Hidden enhancer SIN3A_E3 overlaps with the *CYP1A2* exon, but *CYP1A2* is not expressed in neurons. Indeed, SIN3A-E3 exhibit allelic regulation of *SIN3A* in *cis* (Fig. 3b, Extended Data Fig. 10), exclude the possibility of unexpected effects from genetic deletions.

The E3 in Fig 2c reviewer mentioned is MECP2_E3. It is located in the 3'UTR of *MECP2*, and it overlaps with ATAC-seq, H3K4me1, and H3K27ac peaks (Extended Data Fig. 8). It is not a hidden enhancer. For the CTCF, as shown in Figure 1d, we included CTCF when we checked epigenetic features of the identified enhancers. Therefore, these results suggest that these hidden enhancers are true hidden enhancers do not have typical epigenetic features associated with enhancers. We added the annotation of hidden enhancers in the Supplemental Table 1.

- Please mention in the Main Text whether reduced expression of any of the 4 genes is expected to impact neuronal differentiation and the dynamics of their expression over the course of differentiation. I am wondering how these aspects affect interpretation of the sorting strategy (i.e., are you expecting that some deletions will reduce neural differentiation and therefore lead to indirect effects on genes whose expression is supposed to be up-regulated during differentiation)

We thank the reviewer for raising this important point. Based on previous CRISPRi screening^{6,7} in the same differentiation system, we found that inhibition of *APP*, *FMR1*, and *MECP2* expression did not affect neuronal differentiation, whereas reduced *SIN3A* expression negatively affected neuronal differentiation (Fig. R2). These results are consistent with *SIN3A* being haploinsufficient, while the other three genes are not.

In this study we used tagged reporter lines to monitor target gene expression and sorted neurons with reduced reporter expression. Reduced expression of *APP*, *FMR1*, and *MECP2* does not affect neuronal differentiation^{6,7}. For *SIN3A*, we sorted neurons with reduced expression from one allele, while expression from the other allele maintained *SIN3A* at wildtype level through ACE from the other allele. Therefore, indirect effects due to altered differentiation were avoided in all screens. We have incorporated this information in main text at lines 91-94.

Figure R2. CRISPRi screen results in Tian *et al* ⁷.

Gene	Phenotype	P Value	Gene Score	Hit Class
APP	0.40535852	0.5594594	0.10224416	Non-Hit
FMR1	-0.7898333	0.77813145	-0.08605	Non-Hit
MECP2	-0.8183353	0.38916191	-0.3354108	Non-Hit
SIN3A	-5.7427764	0.00412554	-13.693763	Negative Hit

3. Genetic deletions with Cas9 or 2-guide Cas9 can lead to unexpected outcomes, such as inversions of the deleted sequence, loss of chromosome arms, scars at the cut site that leave a band that appears to be the length of the unedited allele on a PCR gel, etc. Specific questions:

- Have the authors quantified the frequency of such effects in their hiPSC system, and estimated how it might affect the screen results?

We thank the reviewer for bringing up this important question. In our previous study⁸, we checked the performance of dual-sgRNA mediated deletion in H1 hESCs. As shown in Supplementary Figure 2b, a clear band with the expected deletion size was observed, confirming that 2-guide Cas9 system worked efficiently for the intended deletion in hESCs. This dual sgRNA method was also used for other studies, including lncRNA screening⁹. Compared to deletion, the unexpected outcomes or byproducts occur at low efficiency and randomly. Additionally, the unexpected outcomes have limited impact on enhancer screening in our CREST-seq experiments. For example, inversions generally don't affect enhancer activity, while loss of chromosome arms often leads to genome instability and cell death. Other indels are less effective in disrupting enhancer activity compared to deletions.

- What genotyping strategies were used for ensuring such effects were not present in clonal cell lines? E.g., is the 'wild-type' allele in the heterozygous deletion clones indeed unedited?

We thank the reviewer for the question. As shown in Fig. R3, we genotyped the wild-type allele in the heterozygous SIN3A-E4 enhancer (963bp) deletion clones (G-M+). We observed small indels at the sgRNA target site on wild-type alleles. These two sgRNAs (sgRNA1, chr15:74768535-74768554; sgRNA2, chr15:74769843-74769862) targeting DNA sequences outside of SIN3A-E4 (chr15:74768848-74769811) with intended deletion of 1304bp, larger than 963bp SIN3A-E4. The indels happened at the sgRNA target site, which is outside of SIN3A-E4. Therefore, these indels should not affect SIN3A-E4. Indeed, we didn't observe SIN3A downregulation from wild type allele in these heterozygous deletion clones.

Figure R3. The Sanger sequencing results show SIN3A-E4 deletion and indels at sgRNA target sites.

4. Could the authors redefine the phenomenon of “ACE” in the Discussion, and explain more extensively to previously observed phenomena? It seems to me that there are many other similar examples that have been described (e.g., many loci previously found to have negative feedback loops would be expected to behave this way, and heterozygous knockout mice can end up with more than 50% of protein expression). A few specific examples that come to mind include CHD2 (also a chromatin regulator that binds near its own promoter to down-regulate itself) and the many RNA splicing factors

known to auto-regulate themselves e.g. through alternative splicing of poison cryptic exons. It would be helpful to discuss and compare to other such examples.

We thank the reviewer for the valuable suggestions. We added the following to the discussion. “Negative feedback loop (NFL) is a regulatory mechanism that maintains optimized gene dosage across various regulatory layers. At the transcriptional level, for instance, the long noncoding RNA *Chaserr* forms a feedback loop that buffers CHD2 dosage by regulating *Chd2* expression. At the post-transcriptional level, genes autoregulate through a feedback mechanism involving alternative splicing and nonsense-mediated mRNA decay, as seen in *SFPQ*, *RPS3*, and the serine arginine-rich proteins protein family *SRSF1*, *SRSF2*, *SRSF3*, *SRSF4*, *SRSF5*, or *SRSF7*. At the post-translational level, a classic example is the p53–MDM2 loop, in which p53 activates *MDM2* expression, and MDM2 in turn promotes p53 degradation. Compared to these well-characterized NFLs, ACE represents a distinct form of NFL triggered specifically by allelic *cis*-acting enhancer deletion. ACE extends the framework of NFLs by exemplifying a dosage compensation mechanism that is initiated by enhancer loss and mediated through promoter sensing and transcriptional upregulation.” (lines 314-324)

5. Resolution of the analysis

It appears that RELICS sometimes calls very large regions (e.g., 65,981 bp in the gene body of *MECP2*) as a single regulatory element. Why is the resolution at this gene different than others where ‘gene-body enhancers’ are called? Again, viewing the scores for all individual pgRNAs would be helpful here to understand what is going on. The dense tiling (15-20x coverage for each nucleotide) is impressive. Did it pay off? Did it help discover regulatory elements that otherwise would be missed? Commenting on this would strengthen the technical contribution of this paper and motivation for this particular design

We thank the reviewer for these questions. RELICS splits the target region of interest into small segments and checks the impact of each segment on target gene expression. The 65,981 bp region is the *SIN3A* gene body (chr15:75369180-75435161) and annotated as “Gene_body_Exons” instead of an enhancer in the supplemental table 1. It contains 231 functional elements identified by RELICS, and all the elements have the same RELICS score (score = 1). We merged the adjacent functional elements (distance = 0) together in the data analysis. That is why it shows as a larger region. In *APP*, *FMR1*, and *MECP2* gene body regions, we identified individual functional elements. Unlike these regions, we only found one large element in the *SIN3A* region after merging adjacent functional elements. Since this large element overlaps with *SIN3A* exons, we annotated it as “Gene_body_Exons” instead of gene body enhancer.

High throughput methods for enhancer identification includes unbiased^{8,10} and targeted screening¹¹⁻¹³. The method, CREST-seq⁸, we used is an unbiased method which can identify hidden enhancers that lack of classical enhancer marks and would be missed by targeted screening methods. Dual sgRNA mediated deletion can result in different outcomes including deletion, indels at each sgRNA target site, inversion, and retention of wild type allele. Additionally, sgRNAs vary in efficiency. Therefore, using dense sgRNA pairs improves the likelihood of identifying functional enhancers. We added these comments into the discussion part at lines 285-286.

6. Line 203: “the partial reduction of *SIN3A* expression from the promoter deletion allele may not be sufficient to induce ACE...” The data in Fig 4c does not seem to support the hypothesis, esp. since 4b shows compensation happening as early as Day 2, when the knockdown effect for enhancer and promoter seems to be identical (Fig. 4c). Could you explain further?

We thank the reviewer for these comments. As shown in the time course analysis, ACE is a dynamic process. In this process, *SIN3A* downregulation from the deletion allele happened more quickly than upregulation from the compensation allele. On day 2, there was significant downregulation of *SIN3A* (~50%), and early compensation was observed in both enhancer and promoter deletion cells. However, the magnitude of compensation was small, and promoter deletion had a smaller effect size than enhancer deletion. Over time, the observed compensation in promoter deletion cells diminished and eventually disappeared. These data suggest that the compensation observed at day 2 reflects an early, transient response to significant *SIN3A* downregulation. By the final stage of ACE, no compensation occurred in promoter deletion cells. Together, these results support the statement that “the partial reduction of *SIN3A* expression from the promoter deletion allele may not be sufficient to induce ACE.”

Minor points:

- Fig 1b: The layout of the legend was confusing — I thought at first that the colored words corresponded to x-axes or labels for each of the 4 panels, rather than being a legend that applies to all 4 panels. Changing the layout would improve clarity.

We thank the reviewer for their comments. We relabeled the x-axes with their corresponding names in Figure 1b.

- Fig 1b: Filtering to $\log_2FC > 0$ seems inappropriate here. Would it not be better to show the p-value distribution for all pgRNAs?

In these screens, we used FACS to sort out the neurons with reduced expression of reporters. Then, we sequenced the pgRNAs in the sorted cells and control cells to identify the pgRNAs enriched in sorted neurons and the functional elements targeted by the enriched pgRNAs. Therefore, we should focus on the enriched pgRNAs. To follow the reviewer’s suggestion, we added a plot to show the P value distribution for all pgRNAs in the Extended Data Figure 4c. P value indicates the significance of pgRNAs in the sorted cells compared to control cells.

Extended Data Figure 4c. Volcano plots showing the fold changes and P values for each screen. *APP* and *SIN3A* screening data from EGFP and mCherry allele are shown separately.

- Fig 1b: Please define “Test pgRNAs” (blue) in legend

The “Test pgRNAs” in Figure 1b refers to the significantly enriched sgRNAs in the sorted cells with reduced expression of reporters compared with control cells. We relabeled these sgRNAs with “Enriched $P < 0.05$ ”.

- Fig 2a. Please label the site of the reporter insertion knock-in ... how close is it to E3?

We thank the reviewer for their question. We think the reviewer asked Fig 2c instead of Fig 2a, since there is no E3 in Fig 2a. We inserted the EGFP reporter just next to the stop codon of *MECP2*. The EGFP reporter is 4,364bp away from E3.

- “While copy number loss variants overlapped with SIN3a Enhancers identified from clinical samples are likely benign” — Could you clarify whether there is clinical evidence

that the enhancers are benign, or is this an inference from the regulatory model proposed?

We thank the reviewer for this question. We identified two copy number loss variants that overlapped with *SIN3A* enhancers in the ClinVar database (Extended Data Fig. 9d). RCV000139947 is annotated as “likely benign” in ClinVar, consistent with the ACE regulatory model. RCV001834362 is annotated as “Uncertain significance” in the ClinVar database. Based on the ACE regulatory model, we suggest RCV001834362 is less likely contribute to the disease. We have updated the manuscript text to clarify this distinction between clinical annotation and inference from the regulatory model (lines 246-248).

- Generation of deletion clones: Could you clarify how the FACS was done (were clones with low reporter expression single-cell sorted, then genotypes, or was the single-cell sorting + genotyping + clone selection done without regard to reporter expression)? How much time elapsed between single-cell sorting and reporter readout? How many clones were studied in each case (I see two clones in most figures), and how were these two clones selected?

We thank the reviewer for their question. To generate deletion clones, we sorted the single cells with reduced expression of the reporter into 96-well plates with one cell per well. It takes about 3-weeks between single-cell sorting and genotyping. We randomly choose two clones for each analysis.

- Fig. 3d,e — Which enhancer deletion does this represent? Please include in Figure legend, not just the main text.

In Fig. 3d,e, we analyzed the *SIN3A*-E4 enhancer. We added the information in figure legend.

- Line 60: “tilling” → “tiling”

We thank the reviewer for pointing out the typo. We have corrected it.

- Line 83: Is Cas9 expressed the whole time or Cas9 is background silenced after differentiation? So the effective editing time is 1 week or 3 weeks?

We thank the reviewer for bringing up this important question. We delivered constitutive expression of Cas9 and pgRNAs into iPSCs via lentivirus infection. While effective editing time could be long-lasting, most of the editing events happen within a week.

- Line 113: Can you tell this enhancer is a neuronal-specific enhancer from chromatin features such as K27ac?

We thank the reviewer for the question. Based on chromatin markers in neurons from this study and chromatin markers in iPSCs from our previous study¹³, MECP2-E6 enhancer overlapped with ATAC and H3K4me1 peaks in neurons and had weak H3K27ac signal. However, MECP2-E6 did not have an obvious ATAC signal and H3K27ac signal in iPSCs (Extended Data Fig. 8). These results are consistent with our CRISPR deletion experiments, which shows MECP2-E6 is a neuronal-specific enhancer. We have added this information to the manuscript at lines 121-123.

Extended Data Figure 8. Genome browser snapshot of the *MECP2* locus. a, The gray region is the MECP2-E6 deletion region in validation experiment.

- Line 126: What is the orientation of this CTCF motif? Taking into account the relative position of this enhancer to the target gene, does it make sense that disrupting the CTCF affects gene expression?

The orientation of this CTCF motif is reversed (Fig. R4). The distance between this enhancer (SIN3A-E4) (the center of SIN3A-E4, chr15:74769330) and SIN3A TSS (chr15:75,451,690) is 682,361bp. We found 4 CTCF motifs in the *SIN3A* promoter region, but only the left one is located at a CTCF binding peak with the same orientation as the one in SIN3A-E4. Since about 76% loops are between Forward-Reverse CTCF-binding sites (CBS), and about 10% loops are between Forward-Forward or Reverse-Reverse CBS. The loops between Reverse-Forward CBS are very rare (Figure 4A in PMID: 26276636), and we didn't detect significant chromatin interaction between this enhancer and the *SIN3A* promoter using our H3K4me3 PLAC-seq datasets. This enhancer may regulate *SIN3A* through two possible mechanisms. First, SIN3A-E4 might regulate *SIN3A* by using other mechanism, such as TFs linking. Second, based on

Micro-C data generated from WTC11 iPSCs by the 4DN project (<https://data.4dnucleome.org/>, experiment ID: 4DNESODGV2V2), this CTCF motif is located at a TAD boundary (chr15:74765001-74770000) (line 137). Therefore, we propose that deletion of the CTCF motif may alter the TAD structure at this locus and affect *SIN3A* expression.

Figure R4. CTCF motifs at the *SIN3A*-E4 enhancer (a) and the *SIN3A* promoter (b).

- The paragraph on Line 132 about analysis of ClinVar CNVs seems out of place. Perhaps consider moving this to the end of manuscript, or commenting more about the results for the 4 genes under study?

We thank the reviewer for raising this point. The reviewer #1 also pointed out this point. We agree with both reviewers' opinions that the analysis of ClinVar CNVs does not align well with the objectives of this study. Therefore, we decided to remove this part.

- Line 222: Could you remind reader in this paragraph that *SIN3A* is a transcriptional repressor

We thank the reviewer for the comment. We added it to our manuscript (line 227).

- Line 223: The locus plot showing the Sin3A ChIP-seq is helpful. Could you please also comment on how many expected binding sites there are?

We thank the reviewer for the question. According to previous studies, *SIN3A* lacks the ability to bind to chromatin directly. It binds to chromatin through direct interaction with other DNA binding proteins and acts as a master scaffold protein to form a transcriptional complex¹⁴. Therefore, it is not possible to predicting binding sites based on DNA sequences. We leveraged published ChIP-seq datasets (<https://www.encodeproject.org>, access ID: ENCSR000EBO) and identified 6 *SIN3A* binding peaks in the *SIN3A* promoter region (Extended Data Fig. 13b). We validated three of them by ChIP-qPCR (Fig. 5e).

Figure 5e. SIN3A binding at the *SIN3A* promoter region assessed by CHIP-qPCR. Data are shown as mean \pm SD. *P* values were determined using a two-tailed two-sample t-test. Dots represent individual biological replicates. R1: Region 1; R2: Region 2; R3: Region 3.

Extended Data Figure 13b. WashU Epigenome Browser snapshot showing *SIN3A* ChIP-seq signals in H1 cells, and regions checked with CHIP-qPCR.

• Fig. 2G, 3B: From Fig.1, you discovered more than a dozen regulatory elements for *SIN3A*. Could you please clarify why it is plausible that deleting any one of these distal regulatory elements E2-E5, which are more than 600 kb away from the promoter, had such a strong effect on gene expression close to a complete knockdown?

We thank the reviewer for their question. To explore the mechanism of strong regulatory effect of *SIN3A*-E2 - E5, we checked the Micro-C data of WTC11 iPSCs from the 4DN project (<https://data.4dnucleome.org/>, experiment ID: 4DNESODGV2V2). *SIN3A*-E4 is located in a TAD boundary. This is consistent with our result that deleting CTCF motif in *SIN3A*-E4 led to significant downregulation of *SIN3A*. It emphasizes the importance of 3D chromatin structure at this locus in regulating *SIN3A* expression. *SIN3A*-E5 is located just next to the TAD boundary and *SIN3A*-E2 - E5 interact with each other (Fig. R5), suggesting that deleting any of these enhancers may disrupt the 3D chromatin

structure responsible for *SIN3A* expression at this locus, and lead to large *SIN3A* expression change.

Figure R5. The Micro-C contact matrix of SIN3A-E2 - E5 locus.

- Line 237: “Our ACE model can also explain the haploinsufficiency of SIN3A236 for the Witteveen-Kolk syndrome (WITKOS) patients with large deletions of the entire SIN3A locus 237 including the SIN3A promoter...while copy number loss variants overlapped 238 with SIN3A enhancers identified from clinical samples are likely benign.” I’m confused by the logic here – why would losing the whole locus lead to different phenotype as losing the distal enhancer? For the current experiment, the fact that deleting the promoter can’t sufficiently induce ACE can be explained by alternative TSS, but losing the whole locus should surely induce ACE according to your model?

We thank the reviewer for their question. *SIN3A* is a haploinsufficient gene, which means the loss of one copy of *SIN3A* results in cell death. In our ACE model, both copies of *SIN3A* promoter are required to sense downregulated *SIN3A* and initiate ACE. Therefore, losing the entire *SIN3A* locus including its promoter causes haploinsufficiency. In contrast, deleting one copy of enhancer leads to *SIN3A* downregulation, but with both copies of the *SIN3A* promoter remain intact, *SIN3A* downregulation is sensed, and ACE is induced.

- Fig 2J: For this one and the three additional ones in the supplementary figure: the total editing rate is about 50%, 80%, 45%, and 50%. And these are from the cells that have an effect. Then what about the rest of the sequences that don’t have intended edits? Why do they cause a decrease in the expression of the gene?

We thank the reviewer for the question. As shown in Fig 2i, we sorted the cells from EGFP+/mCherry- or EGFP-/mCherry+ population. Then, we extracted genomic DNA from these cells and amplified sgRNA target sites for deep sequencing. Extended Data Figure 12 demonstrates that *SIN3A* enhancer regulates *SIN3A* expression in *cis*. The sorted cells lose *SIN3A* expression from one allele, indicating that mutations only happened on the same allele. Indeed, we found that the editing efficiency is about 50% (Fig.2j, Extended Data Figure 11a middle one and bottom one). One sample showed about 80% editing efficiency (Extended Data Figure 11a upper one), we guess it was caused by PCR bias. Because the shorter DNA fragments (deletions/indels) are easier and faster for DNA polymerase to replicate compared to longer fragments (wild type allele). Overall, the total editing efficiency is about 50% is normal, since deletion/indels only happened on one allele.

- Fig. 4A: Use the diagonal line used in Fig2G to help with visualization

We thank the reviewer for the suggestion. We updated the Fig. 4a.

- Extended Data Figure 6: Why is the enhancer deletion effect biased toward GFP? Not seen for the promoter deletion

We thank the reviewer for the question. We checked the sgRNA target sites using Sanger sequencing and found sequence variations at one of the sgRNA target sites, as shown in the figure below (Fig. R6). These sequence variations can affect sgRNA binding affinity and editing efficiency.

Figure R6. Sanger sequencing results of sgRNA target sites.

Reviewer #3 (Remarks to the Author):

In this manuscript, Ren et al. utilized a high-resolution genetic screening approach to identify functional enhancers of four neuropsychiatric risk genes—APP, FMR1, MECP2, and SIN3A—in iPSC-induced excitatory neurons. Among the several novel enhancers validated through this CRISPR tiling study, the authors focused on the allelic compensation effects (ACE) of targeting SIN3A enhancers. A sophisticated allelic reporter system, tagging SIN3A with EGFP and mCherry on separate alleles, was employed to characterize the transcriptional regulation of each allele upon enhancer deletion.

The results indicate that the loss of SIN3A expression from one allele due to enhancer deletion resulted in increased transcriptional activity on the sister allele, thereby maintaining overall SIN3A expression. The authors propose that this enhancer-mediated ACE is essential for the maintenance of SIN3A—a haploinsufficient gene—and that similar mechanisms might regulate other dosage-sensitive genes.

While the experiments in this study are well-designed and the CRISPR tiling screens provide an unbiased evaluation of the selected gene loci, several issues require further characterization:

1. Fig. 1b: It would be more informative to present the CRISPR screen results using volcano plots, where the x-axis represents log₂ fold-change and the y-axis represents -log₁₀(P value).

We thank the reviewer for the suggestion. We added the volcano plot in the Extended Data Figure 4c.

Extended Data Figure 4c. Volcano plots showing the fold changes and P values for each screen. APP and SIN3A screening data from EGFP and mCherry allele are shown separately.

2. MECP2-E6: MECP2-E6 is characterized as a neuron-specific enhancer. However, the absence of an EGFP-depleted cell population in MECP2-E6-targeted iPSCs (Fig. S5) could also be caused by a lower CRISPR deletion efficiency at the MECP2-E6 locus in iPSCs. The authors should address the limitations of dual-sgRNA deletion efficiency and other alternative possibilities in discussions.

We thank the reviewer for raising this point. To explore MECP2-E6 enhancer, we checked the chromatin markers in neurons from this study and chromatin markers in iPSCs from our previous study¹³. We found MECP2-E6 enhancer overlapped with ATAC and H3K4me1 peaks in neurons and had weak H3K27ac signal. However, MECP2-E6 did not have obvious ATAC signal and H3K27ac signal in iPSCs. These chromatin features suggest MECP2-E6 is a neuron specific enhancer.

Extended Data Figure 8. Genome browser snapshot of the *MECP2* locus. The gray region is the MECP2-E6 deletion region in validation experiment.

3. APP-E3: Targeting APP-E3 (Fig. S6) had a stronger effect on the EGFP allele (7.41% and 5.72%) than on the mCherry allele (0.79% and 0.7%). This unique phenomenon was not observed at other loci. Could the authors explore whether allelic-specific sequences at the APP-E3 locus affect CRISPR targeting efficiency on the mCherry allele?

We thank the reviewer for pointing out this question. We checked the sgRNA target sites of two sgRNAs used for deleting APP-E3 using Sanger sequencing. We found one of the sgRNA target sites contains sequence variations (Fig. R6). Based on the general knowledge of CRISPR/Cas9 system, these variations can affect the sgRNA binding affinity and editing efficiency. The observed sequence variations at one sgRNA target site is the cause of unbalanced editing efficiency at APP-E3 locus.

Figure R6. Sanger sequencing results of sgRNA target sites.

4. Line 197–199: “The observed dynamic rate of ACE after enhancer deletion suggests that ACE is more potent as SIN3A expression approaches the level that triggers haploinsufficiency after day 5.” This observation implies that, in addition to active ACE, survival selection might contribute to the enrichment of higher SIN3A-expressing cells over time. This possibility should be considered.

We thank the reviewer for the comment. We incorporated this possibility in the discussion section (line 332-334).

5. Line 201–204: “Long-read RNA-seq data revealed SIN3A transcription from two TSSs, and we only deleted the promoter of the major SIN3A transcript (Extended Data Fig. 11c,d). Thus, the partial reduction of SIN3A expression from the promoter deletion allele may not be sufficient to induce ACE.” This hypothesis should be tested by deleting both SIN3A TSSs to determine whether ACE is observed under complete loss of SIN3A expression from one allele.

We thank the reviewer for raising this point. In this revision, we designed two sgRNA pairs and performed deletion experiments to delete both *SIN3A* TSSs. We infected *SIN3A* reporter cells with lentivirus expressing sgRNAs and Cas9. After puromycin selection. We observed down-regulation of *SIN3A* expression from sgRNA pair #1 and #2 without an obvious compensation effect. These results suggest that promoter deletion is not sufficient to induce ACE (Extended Data Fig. 13c).

Extended Data Figure 13c. Flow cytometry plots showing the EGFP and mCherry signals in control cells (WTC11 ⁱ³N), SIN3A-EGFP/mCherry reporter cells, and SIN3A promoter deletion cells.

6. Line 218–219: “These results suggest ACE, once established, cannot be reversed by increasing SIN3A expression.” The ectopic expression of SIN3A-Pr > SIN3A-P2A-BFP failed to reverse SIN3A enhancer ACE (Fig. 4f), raising concerns about whether the “dosage effect” is the primary contributor to ACE. To address this, overexpressing SIN3A using a viral promoter (e.g., CMV) might bypass potential autoregulatory feedback affecting SIN3A-Pr > SIN3A-P2A-BFP expression. Additionally, the authors should compare the effects of ectopic expression of SIN3A-P2A-BFP before versus after SIN3A enhancer deletion to clarify the proposed hypothesis.

We thank the reviewer for raising this point. To abolish the *SIN3A* autoregulatory feedback, we cloned the CAG promoter controlled SIN3A-P2A-BFP and delivered it into control cells (wild type, SIN3A-EGFP/mCherry) and SIN3A-E4 deletion clones via lentivirus infection. First, we performed RT-qPCR to confirm the overexpression of *SIN3A* in the lentivirus-infected cells (Extended Data Fig. 13d). Compared to about 1.7-fold SIN3A overexpression (Extended Data Figure 13e), CAG promoter resulted in more than 3-fold expression of SIN3A compared to the endogenous expression level. Next, we compared SIN3A-E4 clones with and without CAG>SIN3A-P2A-BFP and found that the *SIN3A* overexpression did not affect the SIN3A ACE (Fig. 4g).

Extended Data Figure 13d,e. RT-qPCR showing overexpression of *SIN3A* controlled by the CAG or SIN3A promoter.

Figure 4g. Flow cytometry plots showing the SIN3A-EGFP and SIN3A-mCherry signals in control cells and SIN3A-E4 deletion clones with and without CAG promoter controlled ectopic *SIN3A* expression. SIN3A-EGFP/mCherry reporter cells were used as control.

Finally, we deleted the SIN3A-E4 in CAG>SIN3A-P2A-BFP cells and isolated individual clones, and we observed the compensation in both wild type and SIN3A overexpression conditions. To quantitatively compare the compensation effect, we calculated the compensation index (CI) defined as the ratio of EGFP signal or mCherry signal (from the compensation allele) in enhancer deletion clones relative to control cells. We found

a significant lower CI in CAG>SIN3A-P2A-BFP background (P values are <0.05), which suggests *SIN3A* overexpression suppresses enhancer deletion induced ACE (Fig. 4f).

Figure 4f. *SIN3A* overexpression represses ACE. The ACE in *SIN3A*-E4 deletion clones isolated from *SIN3A*-EGFP/mCherry reporter cells with and without CAG promoter controlled ectopic *SIN3A* expression. C1 and C2 indicate two independent clones for each genotype. Control cells are *SIN3A*-EGFP/mCherry reporter cells, with (CAG>*SIN3A*) or without *SIN3A* overexpression (WT). The compensation index (CI) is defined as the ratio of EGFP or mCherry signal (from the compensation allele) in enhancer deletion clones relative to control cells. P values were determined using the two-tailed two-sample t-test.

7. ChIP-qPCR: To support the proposed model summarized in Fig. 5e, ChIP-qPCR analysis of *SIN3A* binding at *SIN3A* promoter loci should be performed.

We thank the reviewer for the suggestion. We performed ChIP-qPCR in both iPSCs and 2-week neurons. Besides two promoter regions P1 (Region1) and P2 (Region2), we included two additional regions near the *SIN3A* promoter region, one region without *SIN3A* binding signal as negative control (NegCtrl) and one *SIN3A* ChIP-seq peak (Region3). We observed significant enrichment of the *SIN3A* binding signal at the Region 1, Region 2 and Region 3 compared with the negative control region. These results prove the binding of *SIN3A* at the promoter regions, P1 and P2, that we tested. The results are added in the Figure 5e and Extended Data Figure 13b.

Figure 5e. SIN3A binding at the SIN3A promoter region assessed by ChIP-qPCR. Data are shown as mean \pm SD. P values were determined using a two-tailed two-sample t-test. Dots represent individual biological replicates. R1: Region 1; R2: Region 2; R3: Region 3.

Extended Data Figure 13b. WashU Epigenome Browser snapshot showing *SIN3A* binding regions checked with ChIP-qPCR.

Reference:

1. Li, Y. E. *et al.* A comparative atlas of single-cell chromatin accessibility in the human brain. *Science* **382**, eadf7044 (2023).
2. Emani, P. S. *et al.* Single-cell genomics and regulatory networks for 388 human brains. *Science* **384**, eadi5199 (2024).
3. Zemke, N. R. *et al.* Conserved and divergent gene regulatory programs of the mammalian neocortex. *Nature* **624**, 390–402 (2023).
4. Yan, J. *et al.* Discovery of NANOG enhancers and their essential roles in self-renewal and differentiation in human embryonic stem cells. *Stem Cell Rep.* **20**, 102511 (2025).
5. Doench, J. G. *et al.* Optimized sgRNA design to maximize activity and minimize off-target effects of CRISPR-Cas9. *Nat. Biotechnol.* **34**, 184–191 (2016).
6. Yang, X. *et al.* Functional characterization of gene regulatory elements and neuropsychiatric disease-associated risk loci in iPSCs and iPSC-derived neurons. Preprint at <https://doi.org/10.1101/2023.08.30.555359> (2023).
7. Tian, R. *et al.* Genome-wide CRISPRi/a screens in human neurons link lysosomal failure to ferroptosis. *Nat. Neurosci.* **24**, 1020–1034 (2021).
8. Diao, Y. *et al.* A tiling-deletion-based genetic screen for cis-regulatory element identification in mammalian cells. *Nat. Methods* **14**, 629–635 (2017).
9. Zhu, S. *et al.* Genome-scale deletion screening of human long non-coding RNAs using a paired-guide RNA CRISPR-Cas9 library. *Nat. Biotechnol.* **34**, 1279–1286 (2016).
10. Fulco, C. P. *et al.* Systematic mapping of functional enhancer-promoter connections with CRISPR interference. *Science* **354**, 769–773 (2016).

11. Klann, T. S. *et al.* CRISPR-Cas9 epigenome editing enables high-throughput screening for functional regulatory elements in the human genome. *Nat. Biotechnol.* **35**, 561–568 (2017).
12. Diao, Y. *et al.* A new class of temporarily phenotypic enhancers identified by CRISPR/Cas9-mediated genetic screening. *Genome Res.* **26**, 397–405 (2016).
13. Ren, X. *et al.* Parallel characterization of cis-regulatory elements for multiple genes using CRISPRpath. *Sci. Adv.* **7**, eabi4360 (2021).
14. Grzenda, A., Lomberk, G., Zhang, J.-S. & Urrutia, R. Sin3: master scaffold and transcriptional corepressor. *Biochim. Biophys. Acta* **1789**, 443–450 (2009).

REVIEWER COMMENTS

Reviewer #1 (Remarks to the Author):

I would like to thank the authors for addressing my comments carefully. The only response I find a bit worrying is in response to my initial comment on the previous lines 91-104 where I asked for a comparison to previously established regulatory elements. I proposed there to use a human specific atlas of such regulatory elements generated by a large meta-analysis of human brain epigenome data. The authors reply that they did not use this resource for their comparisons, as it would contain data sets that would not be derived from human samples. This is incorrect (only some human regulatory elements in that study have been tested in animal models, but are still human sequences). Hence this incorrect reply makes me suspiciously wonder whether the authors might have tried to use that alternatively proposed atlas and might have found results that did not fit into their expectations and might have therefore chosen alternative sources where the results might have fit their expectations better? The authors might want to comment on that, to prevent the impression of any potential selection bias introduced here.

We thank Reviewer #1 for the comments. In our previous response, we noted that some of the datasets in PMID: 34663447, including data from cell lines such as SK_N_MC and SK_N_SH, or from hiPSC/ESC-derived cells, are not derived from primary cells. Consequently, our previous analysis only used data generated from cells directly isolated from human primary tissues. After carefully considering the reviewer's comments, we agree that it is appropriate to include the datasets in PMID: 34663447 in our analysis, since the excitatory neurons used in our screens were differentiated from hiPSCs rather than primary human tissues. We compared our identified enhancers with the datasets from PMID: 34663447 and found that canonical enhancers showed a significantly higher overlap with human brain cCREs compared with hidden enhancers (100% vs 63.6%, $P = 0.004$, Fisher's exact test). This result is consistent with the findings from our previous analysis. We added this new information to the manuscript (lines 108-109 and Figure 1e).

	Overlap with cCREs generated in PMID: 34663447
Canonical enhancers	28/28=100%
Hidden enhancers	7/11=63.6%

Reviewer #2 (Remarks to the Author):

The reviewers have revised the manuscript substantially and made numerous improvements.

A few additional edits would be sufficient to address my previous major points:

- For the Extended Data Figure 4 (reproducibility of CREST-seq screens), please show full scatterplot of RELICS probability scores between replicates 1 and 2 for each screen (e.g., each point = one segment, 'hit' segments colored) rather than simply reporting the Pearson correlation, because good Pearson correlations can be achieved by e.g. single outliers. Also please report the Pearson correlation of RELICS probability scores considering just the 'hits'. I also think it is necessary to show scatterplots at the level of individual pgRNAs, not just a PCA plot.

We thank Reviewer #2 for the comment and suggestion. First, we added scatterplots for the individual pgRNAs in each screen (Extended Data Figure 3c). We demonstrate that the positive control pgRNAs (green and red dots) exhibited higher correlations than the controls, confirming that our experimental strategy successfully identified functional DNA elements responsible for target gene expression. We observed a relatively low correlation in the sorted samples, characterized by reduced reporter expression.

Extended Data Figure 3c. Scatter plots showing the distribution of individual pgRNAs in each screen. The Pearson correlation coefficient (PCC) values for all pgRNAs (Total) and positive control pgRNAs (EGFP and mCherry pgRNAs: GM) are indicated in each plot.

Second, according to the reviewer's suggestion, we added scatterplots of RELICS probability scores between replicate 1 and 2 for each screen, as well as between the EGFP and mCherry alleles for the APP and SIN3A screens (Extended Data Figure 4b).

As shown in the figure below, we calculated the Pearson correlation for all segments, hit segments, and non-hit segments. Overall, we observed relatively high correlations for all segments and hit segments, and low correlations for non-hit segments across all screens.

Extended Data Figure 4b. Scatterplots of RELICS probability scores. The scores are shown for all segments, hit segments, and non-hit segments. The Pearson correlation coefficient (PCC) values and linear regression lines are indicated in each plot.

Interestingly, we observed variations at the pgRNA and segment level for a subset of our data, and would like to provide some potential explanations here. In our screens, we dissociated 2-week neurons with Papain and sorted ~1% of cells that exhibited the lowest reporter expression (Extended Data Figure 3a). In total, we collected about 500,000 cells per replicate. Assuming that ~10% of pgRNAs in each library affect target gene expression, the coverage of the sorted cells would be approximately 300-500 \times . However, when considering all pgRNAs, the coverage is less than 50 \times . We think that the observed variations may be due to this low coverage. Since RELICS will consider data from both replicates for calling significant hits, and our extensive validation experiments have confirmed that the identified enhancers are functional, even though they may show differences in RELICS scores across individual replicates. Nevertheless, we suspect some weaker enhancers for these four genes may have been missed due to our stringent cell sorting strategy. To highlight the limitations of this study and provide guidance for future use of this method, we included the following in the Discussion section (lines 295-300).

“At the individual pgRNA level and DNA fragment level, we observed some variation between replicates. We suspect this variation stems from the low number of cells collected from the sorted samples (approximately the bottom 1% of cells with reduced expression), which may have led to the exclusion of weaker enhancers in our screens. We advise increasing the number of collected cells and incorporating additional replicates when feasible to enhance the robustness of results when using the CREST-seq method.”

- Thanks for adding the genome browser plots, very helpful. One small addition that would make these even better would be to show in the browser track which regions are covered by pgRNAs (are there any gaps in coverage due to repetitive sequences or other design choices/limitations?)

We thank Reviewer #2 for the suggestion. As shown in the Extended Data Figure 2b, there are no gaps in any of the target regions. To clarify this, we added the pgRNA tracks to the genome browser session, allowing readers to zoom in and examine the pgRNA library in detail (https://epigenomegateway.wustl.edu/browser2022/?genome=hg38&sessionFile=https://shen-xren.s3.us-west-1.amazonaws.com/CREST-seq_NC/eg-session-YzGAc9x4n-ce92ef10-72ad-11f0-b5d2-a94bbd38dced.json).

We also updated the Extended Data Figure 6 by adding the pgRNA track for each locus. Overall, these data illustrations consistently demonstrate that there are no gaps in the target regions.

- In the Response, the authors wrote: “In our screens, we identified functional elements based on their impact on reduced target gene expression when perturbed. Because these elements positively regulate transcription, we believe the term “enhancer” is appropriate for describing the functional elements we identified.” I disagree with this definition, but, at the very least, the authors should include their own definition early in the manuscript.

We thank Reviewer #2 for the suggestion. We added this information to the manuscript at lines 96-97.

- This sentence of the response should be included in the Methods or Supp Figure legend, because it is a critical detail for interpreting the expression effects observed in the deletion clones: “To generate deletion clones, we sorted the single cells with reduced expression of the reporter into 96-well plates with one cell per well. It takes

about 3-weeks between single-cell sorting and genotyping. We randomly choose two clones for each analysis.”

We thank Reviewer #2 for the suggestion. We added this information to the Methods part of the manuscript at lines 610-612.

Reviewer #3 (Remarks to the Author):

The authors have addressed the previous concerns by including additional data and analyses, which have considerably strengthened their claims. The manuscript is now substantially improved, and I have no further comments.

We thank Reviewer #3 for their constructive comments, which helped us improve our study.

Reviewer #4 (Remarks to the Author):

We thank Reviewer #3 for their constructive comments, which helped us improve our study.